# Coordinate regulation of mutant NPC1 degradation by selective ER autophagy and MARCH6-dependent ERAD

Mark L. Schultz[1], Kelsey L. Krus[1], Susmita Kaushik[2], Derek Dang[1], Ravi Chopra[3,4], Ling Qi [5], Vikram G. Shakkottai[4,5], Ana Maria Cuervo [2] & Andrew P. Lieberman[1]

Niemann–Pick type C disease is a fatal, progressive neurodegenerative disorder caused by loss-of-function mutations in NPC1, a multipass transmembrane glycoprotein essential for intracellular lipid trafficking. We sought to define the cellular machinery controlling degradation of the most common disease-causing mutant, I1061T NPC1. We show that this mutant is degraded, in part, by the proteasome following MARCH6-dependent ERAD. Unexpectedly, we demonstrate that I1061T NPC1 is also degraded by a recently described autophagic pathway called selective ER autophagy (ER-phagy). We establish the importance of ER-phagy both in vitro and in vivo, and identify I1061T as a misfolded endogenous substrate for this FAM134B-dependent process. Subcellular fractionation of I1061T *Npc1* mouse tissues and analysis of human samples show alterations of key components of ER-phagy, including FAM134B. Our data establish that I1061T NPC1 is recognized in the ER and degraded by two different pathways that function in a complementary fashion to regulate protein turnover.

[1] Department of Pathology, University of Michigan School of Medicine, Ann Arbor, MI 48109, USA. [2] Department of Developmental and Molecular Biology, Albert Einstein College of Medicine, Bronx, NY 10461, USA. [3] Medical Scientist Training Program, University of Michigan Medical School, Ann Arbor, MI 48109, USA. [4] Department of Neurology, University of Michigan Medical School, Ann Arbor, MI 48109, USA. [5] Department of Molecular & Integrative Physiology, University of Michigan, Ann Arbor, MI 48109, USA. Correspondence and requests for materials should be addressed to A.P.L.(email: liebermn@umich.edu)

The biosynthesis of transmembrane glycoproteins initiates in the endoplasmic reticulum (ER), a site where native folding and initial post-translational modifications occur. Protein folding is guided by the resident quality control machinery, which facilitates and regulates complex steps underlying co-translational glycosylation, chaperone-assisted folding, and regulated export from the ER[1,2]. This multi-step process is inherently error-prone, and misfolded proteins are either aberrantly retained within the ER or targeted for degradation. The importance of ER quality control to human health is underscored by the occurrence of missense mutations in multiple genes directly linked to disease, resulting in loss-of-function because of ER retention or degradation of misfolded, mutant proteins.

Among diseases caused by mutations that impair folding of transmembrane glycoproteins is Niemann–Pick type C disease, a fatal and progressive neurodegenerative disorder characterized by the intracellular accumulation of unesterified cholesterol[3]. Although symptom onset and disease severity are variable, patients often develop hepatosplenomegaly, progressive cognitive decline, seizures, and death before age 30[4,5].

The vast majority (~95%) of Niemann–Pick type C patients harbor mutations in the gene encoding NPC1, a structurally complex 13 transmembrane domain glycoprotein. NPC1 is synthesized in the ER, traffics through the Golgi where its glycans are modified, and resides in the late endosomal/lysosomal (LE/Lys) compartment[6]. Crystal and cryo-EM structures confirm that NPC2, a soluble protein in the lumen of LE/Lys[7,8], hands unesterified cholesterol to NPC1 for insertion into the lysosomal membrane. This insertion event is required for cells to access LDL-derived cholesterol for use in membranes or steroid hormone production[9].

Approximately 250 different loss-of-function mutations in the NPC1 gene have been identified as causative of disease. The most common mutation is an isoleucine to threonine missense mutation at position 1061 (I1061T), found in 20% of patients of western European descent[10]. By studying endogenous and over-expressed I1061T, previous studies found that this mutant is recognized in the ER and rapidly degraded by the proteasome[11,12]. Importantly, overexpression of I1061T or the ER chaperone calnexin facilitates its trafficking from the ER to the LE/Lys compartment, where it is functional[12,13]. This has prompted ongoing investigations of therapies which modulate cellular proteostasis pathways (reviewed in ref. [14]). However, little is known about the machinery that recognizes and regulates the degradation of misfolded NPC1 mutants, including I1061T.

Here, we describe that the endogenous I1061T mutant is recognized in the ER and degraded by two independent pathways. A portion of I1061T is recognized by MARCH6-dependent endoplasmic-reticulum-associated degradation (ERAD) and targeted to the proteasome. Alternatively, a substantial portion of I1061T is recognized by the recently described autophagic pathway called selective ER autophagy (ER-phagy). We identify I1061T NPC1 as an endogenous misfolded substrate degraded by this FAM134B-dependent process and demonstrate the importance of this pathway both in vitro and in vivo. Subcellular fractionation of I1061T Npc1 mouse tissues and western blotting of human samples show alterations of key components of ER-phagy, including the critical receptor protein FAM134B. These data establish that I1061T NPC1 is recognized in the ER and degraded by two different pathways, which function in a complementary fashion to regulate protein turnover.

## Results

**I1061T NPC1 accumulates after lysosomal or proteasomal inhibition.** The I1061T protein is recognized in the ER and

degraded with a half-life of approximately 6.5 h, while the wild-type (WT) protein is degraded with a half-life approximating of 9 h; misfolded species of both proteins likely turnover more rapidly[12]. To begin defining the mechanisms of I1061T degradation, we analyzed lysates from control (CTRL) cells expressing WT NPC1 and I1061T homozygous primary fibroblasts after treatment with the proteasome inhibitors MG132 or epoxomicin (Epox) at non-toxic concentrations (Supplementary Fig. 1a). Consistent with previous studies[11,12], proteasome inhibition caused a significant increase in WT and I1061T protein levels (Fig. 1a). Relative to WT, proteasome inhibition of cells expressing I1061T recovered only ~40% of the degraded protein, raising the possibility that I1061T was concurrently degraded by proteasome-independent mechanisms.

To determine whether I1061T was degraded in the lysosome, CTRL and I1061T cells were treated with Bafilomycin A1 (Baf). Baf inhibits lysosomal acidification by the vacuolar H$^+$-ATPase. WT NPC1 protein levels significantly increased following treatment with Baf (Fig. 1a), consistent with its stabilization within the lysosome. Unexpectedly, Baf treatment also recovered I1061T protein to WT levels (Fig. 1a) indicating that the lysosome is a major compartment utilized in I1061T degradation. Consistent with previous reports[11–13], treatment with the lysosomal inhibitor chloroquine did not significantly alter I1061T protein levels (Supplementary Fig. 1b). While this difference likely reflects an increased efficacy of Baf to neutralize lysosomal pH relative to chloroquine, other possibilities cannot be excluded.

In contrast to WT NPC1, which was degraded to a similar extent in the proteasome and lysosome, I1061T was predominantly degraded in the lysosome (Fig. 1b). A time course of Baf treatment in CTRL fibroblasts revealed an accumulation of higher molecular weight WT NPC1 (Fig. 1c), likely indicating that inhibition of lysosomal acidification interfered with the turnover of fully glycosylated species. In contrast, no size shift was observed with the more modest accumulation of I1061T protein (Fig. 1c). In cells expressing I1061T, concurrent inhibition of both the proteasome and lysosome (Baf+Epox) had additive effects on NPC1 protein levels at early time points (Supplementary Fig. 1c). Collectively, these studies suggested that the I1061T mutant was degraded by both the proteasome and lysosome.

We sought to corroborate our findings in an independent cell line expressing I1061T NPC1. Moreover, little is known about the degradation pathways regulating many of the other disease-causing mutants[15]. NPC1 compound heterozygous primary patient fibroblasts (C.1947+5G>C/I1061T, P1007A/T1036M) were treated with MG132 or Baf to determine the relative contributions of proteasomal- and lysosomal-based degradation (Fig. 1d). We additionally studied two other compound heterozygous mutants (Supplementary Fig. 1d). Like cells homozygous for I1061T NPC1, these lines showed low baseline levels (5–34%) of mutant NPC1 protein relative to WT. Inhibition of the lysosome by Baf significantly increased NPC1 levels in all four compound heterozygous cell lines (Fig. 1d, Supplementary Fig. 1d.). In contrast, inhibition of the proteasome significantly increased levels only in the P1007A/T1036M and C.1947+5G>C/I1061T mutants. We conclude that several NPC1 missense mutants, including I1061T, are degraded by both the lysosome and proteasome. Of interest is the P1007A/T1036M mutant, which exhibits the greatest accumulation (~10 fold) compared to WT NPC1 (~2.5 fold) following Baf treatment (Fig. 1d).

**I1061T NPC1 is degraded in part by MARCH6-dependent ERAD.** The accumulation of I1061T protein following inhibition

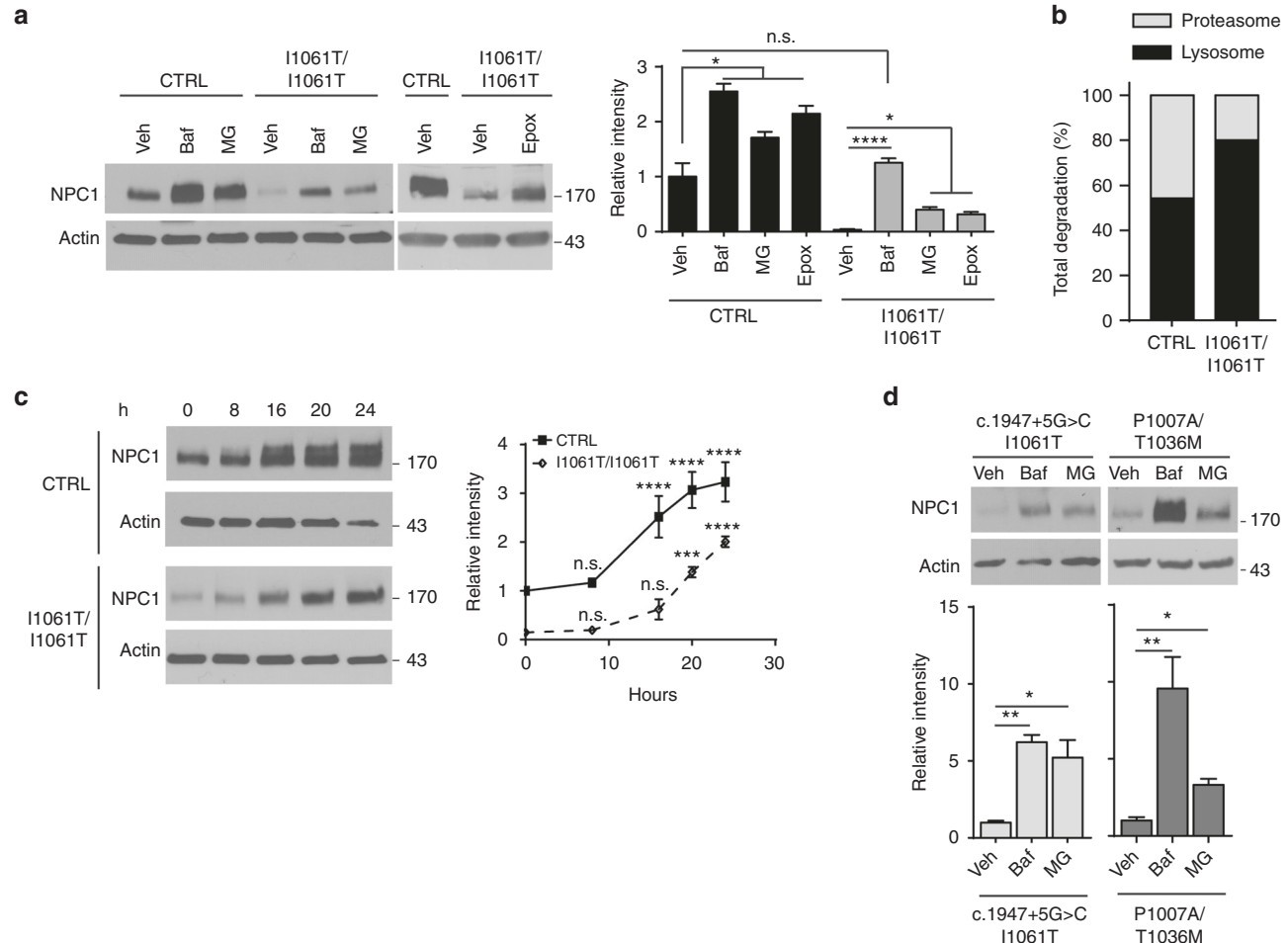

**Fig. 1** I1061T NPC1 accumulates after lysosomal or proteasomal inhibition. **a** Primary human fibroblasts homozygous for WT *NPC1* (CTRL) or I1061T *NPC1* (I1061T) were treated with vehicle (Veh),100 nM bafilomycin A1 (Baf), 10 μM MG132 (MG), or 100 nM epoxomicin (Epox) for 24 h. Quantified at right. **b** NPC1 protein levels quantified in **a** were normalized to calculate relative contribution of each pathway to NPC1 degradation. **c** Time course of NPC1 accumulation after treatment with Baf. Quantified at right. **d** Patient fibroblasts with the indicated *NPC1* mutations were treated for 24 h with vehicle, 100 nM Baf, or 10 μM MG. Normalized relative to CTRL and quantified below. **a**–**d** Data are mean ± s.e.m. from three independent experiments. n.s., not significant, $*P \leq 0.05$, $**P \leq 0.01$, $***P \leq 0.001$, $****P \leq 0.0001$ by ANOVA with **a**–**d** Tukey's or **c** Bonferroni posthoc test (**a** $F = 58.71$; **c** $F = 7.68$; **d** $F = $ (c.1947+5G>C/I1061T) 14.91, (P1007A/T1036M) 13.46)

of the proteasome is consistent with previous studies[11,12] and suggests that the I1061T protein is triaged through ERAD. We sought to define the ERAD machinery involved in this process.

The E3 ubiquitin ligases HRD1, GP78, and MARCH6 contribute to the degradation of many ERAD substrates[16]. For functional activity, HRD1 utilizes the adaptor proteins Sel1L[17] and Derlin-1[18]. To initially test the involvement of this machinery in NPC1 degradation, we used mouse embryonic fibroblasts homozygous for a conditional *Sel1L* null allele[17]. Tamoxifen treatment efficiently eliminated Sel1L expression in these cells through the action of hormone-regulated Cre recombinase (Fig. 2a). The resulting deficiency of Sel1L did not alter the extent to which WT NPC1 protein accumulated following treatment with MG132 (Fig. 2a), suggesting that this machinery did not play a significant role in shuttling NPC1 to the proteasome. Similarly, knockdown or over-expression of Derlin-1, HRD1, or GP78 did not alter I1061T protein levels (Fig. 2b, c).

The lack of involvement of the E3s HRD1 and GP78 in I1061T degradation prompted us to investigate the contribution of MARCH6. MARCH6 is an E3 ubiquitin ligase that regulates the degradation of multiple proteins in the cholesterol metabolic and biosynthetic pathways[19–21]. Knockdown of MARCH6 significantly

increased I1061T protein levels without altering WT NPC1 levels (Fig. 2d, Supplementary Fig. 2a), consistent with the involvement of MARCH6-dependent ERAD in the degradation of I1061T NPC1.

After substrate ubiquitination, the AAA ATPase p97 recognizes and extracts misfolded substrates into the cytoplasm for degradation by the proteasome[22]. Treatment of CTRL and I1061T cells with the p97 inhibitor eeyarestatin I (Eey) resulted in the accumulation of both ubiquitinated proteins (Supplementary Fig. 2b) and high molecular weight NPC1 species (Fig. 2e). This robust effect of p97 inhibition on NPC1 levels likely reflected its critical role in both ERAD and autophagy[23–25].

**I1061T traffics from the ER to autophagosomes**. The observation that inhibition of either the proteasome or lysosome increased levels of NPC1 missense mutants prompted us to determine the extent to which these accumulating proteins were functional. To accomplish this, we stained cells with the fluorescent dye filipin to label unesterified cholesterol. Although both MG132 and Baf increased I1061T protein levels (Fig. 1a), neither one rescued cholesterol accumulation

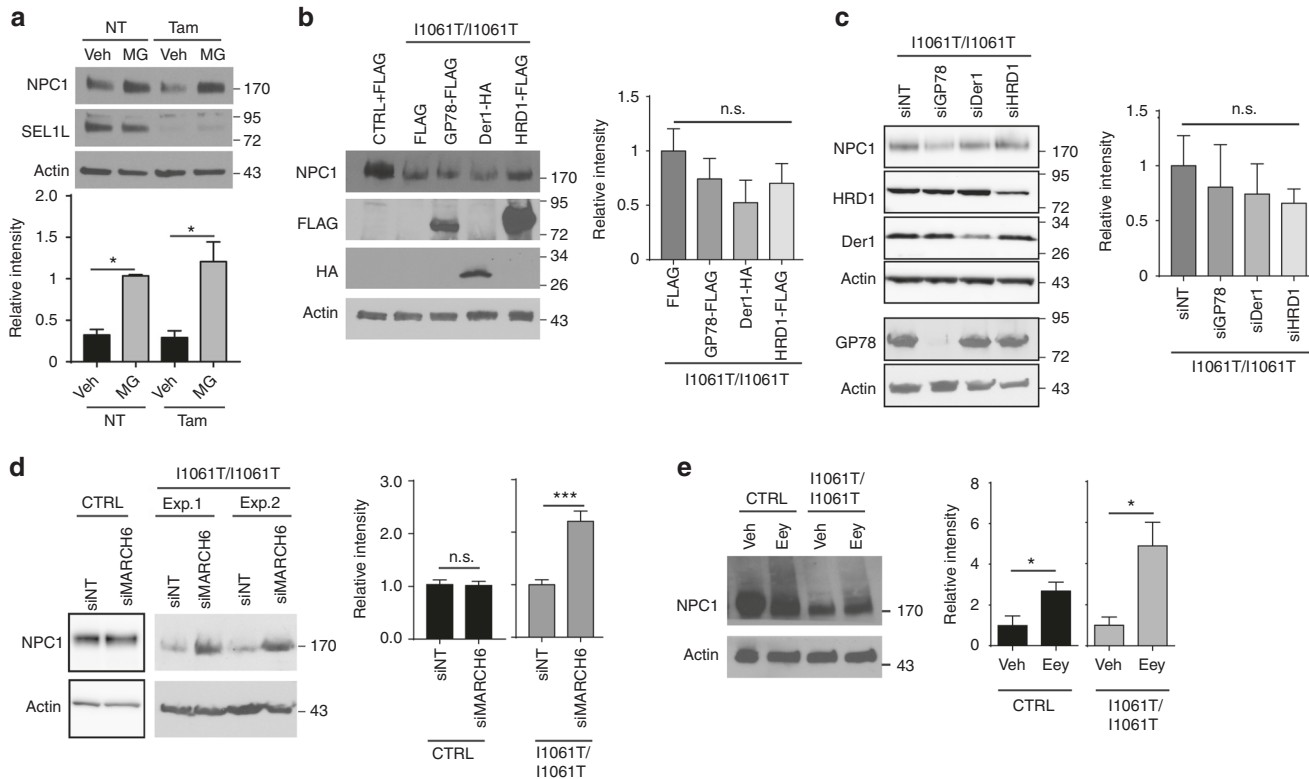

**Fig. 2** I1061T NPC1 is degraded in part by MARCH6-dependent ERAD. **a** Tamoxifen-inducible *Sel1L* null mouse embryonic fibroblasts expressing WT *NPC1* were treated with vehicle (NT) or 200 nM tamoxifen (Tam) for 48 h. Cells were also treated with vehicle (Veh) or 10 μM MG132 (MG) during the last 24 h to determine if Sel-1L deficiency alters the pool of WT NPC1 trafficking to the proteasome. NPC1 protein level quantified below. **b** Over-expression of FLAG, GP78-FLAG, Der1-HA, and HRD1-FLAG in I1061T *NPC1* fibroblasts. Cells were transfected at $t = 0$ and $t = 24$ h and lysates collected at 48 h. Transgene expression confirmed using antibodies against FLAG and HA. NPC1 quantified at right, normalized to I1061T *NPC1* fibroblasts transfected with FLAG. **c** I1061T *NPC1* fibroblasts were treated with the following siRNAs: non-targeting (NT), GP78, Der1, and HRD1 at $t = 0$ and $t = 24$ h. Lysates were collected at 48 h. NPC1 quantified at right, normalized to I1061T *NPC1* fibroblasts transfected with NT siRNA. **d, e** CTRL and I1061T *NPC1* fibroblasts were treated with **d** non-targeting or MARCH6 siRNA (shown are two independent experiments for I1061T), **e** vehicle (Veh) or 10 μM eeyarestatin I (Eey). Lysates were blotted for NPC1 (quantified at right). Data are mean ± s.e.m. **a–e** $N = 3$–4, **b**, **c** $N = 5$–6, **d** $N = 4$–6. **a–c** One-way ANOVA with Tukey's posthoc test, **a** $F = 8.147$. **d**, **e** Student's $t$-test, **d** (I10) $t = 4.8$, **e** $t =$ (CTRL) 2.69, (I10) 3.16. n.s. = not significant, $*P \le 0.05$, $***P \le 0.001$

(Fig. 3a). In contrast, cyclodextrin, which is known to reduce cellular cholesterol, significantly rescued storage (Fig. 3a). These data indicated that the accumulating I1061T protein was non-functional.

To further analyze trafficking of WT and missense mutants, we took advantage of the fact that glycans are added to NPC1 in the ER. Upon trafficking to the medial Golgi, these glycans are modified so that they are resistant to digestion by endoglycosidase H (EndoH), but remain sensitive to peptide-N-glycosidase F (PNGase)[26]. Enzymatic removal of glycans enhanced the mobility of NPC1 on SDS-PAGE. Consistent with the previous data[12,26,27], WT NPC1 was resistant to EndoH digestion (Fig. 3b), indicating that most cellular proteins had trafficked through the medial Golgi. A similar pattern was seen with WT NPC1 after treatment with Baf. Treatment of CTRL fibroblasts with epoxomicin accumulated an EndoH sensitive species, likely reflecting a pool of misfolded WT protein that is normally trafficked through ERAD.

In contrast, I1061T was sensitive to EndoH digestion (Fig. 3b), indicating a failure to traffic through the medial Golgi[12]. After treatment with Baf or epoxomicin, the accumulating I1061T protein remained sensitive to EndoH (Fig. 3b). Notably, trafficking through the Golgi was not rate limiting for I1061T protein degradation since treatment with Brefeldin A (10 μg/ml), an inhibitor of ER to Golgi trafficking,

did not significantly increase I1061T protein levels (Supplementary Fig. 1b). Similarly, most I1061T protein that accumulated after MARCH6 knockdown was sensitive to EndoH and was not functionally active (Supplementary Fig. 2c, d). We conclude that these accumulating species did not advance through the secretory pathway, but instead trafficked directly from the ER to the proteasome or lysosome for degradation.

An interesting contrast was observed in cells expressing P1007A/T1036M NPC1. Treatment with Baf, but not MG132, significantly reduced cholesterol storage in these cells (Fig. 3a). This raised the possibility that a fraction of mutant NPC1 protein trafficked to the lysosome, where it was stabilized by Baf to partially rescue its function. This notion was supported by the observation that, at baseline, P1007A/T1036M fibroblasts expressed a significant fraction of NPC1 protein that was EndoH resistant (Fig. 3b). Treatment with Baf accumulated both EndoH sensitive and resistant species (Figs. 1c and 3b). In contrast, treatment with proteasome inhibitors had more modest effects on total NPC1 protein levels (Fig. 1c) and function (Fig. 3a), and favored the accumulation of EndoH sensitive species (Fig. 3b). These analyses indicated that in contrast to the I1061T mutant, P1007A/T1036M NPC1 trafficked, albeit inefficiently, to late endosomes/lysosomes where it displayed diminished function.

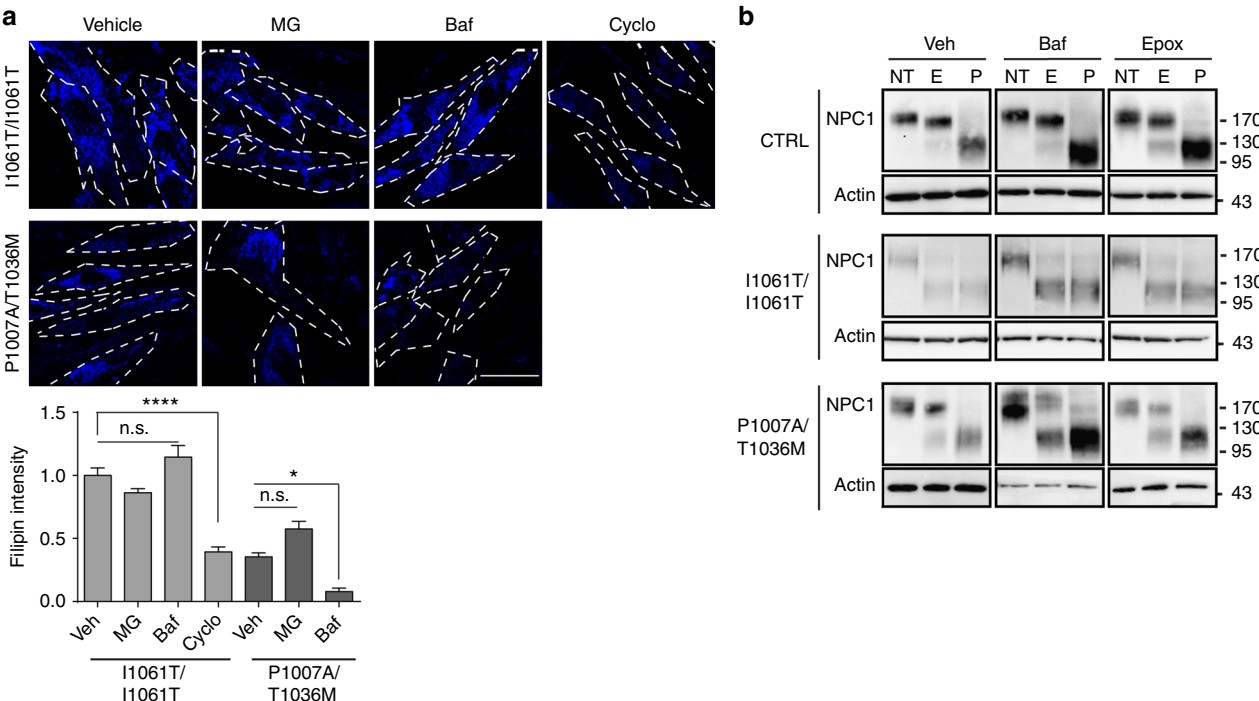

**Fig. 3** Accumulating I1061T NPC1 does not traffic to lysosomes. **a** Primary human fibroblasts expressing I1061T or P1007A/T1036M *NPC1* were treated for 24 h with vehicle, 10 μM MG132 (MG), 100 nM bafilomycin A1 (Baf), or 1 mM cyclodextrin (Cyclo). Unesterified cholesterol was labeled with filipin (blue), and staining intensity calculated from three independent experiments (16 fields/experiment). Scale bar = 50 μm. Dashed lines outline plasma membrane. Quantified at the bottom. **b** CTRL, I1061T, or P1007A/T1036M *NPC1* fibroblasts were treated for 24 h with vehicle (Veh), 100 nM bafilomycin A1 (Baf), or 100 nM epoxomicin (Epox). Lysates were digested with endoglycosidase H (E), PNGaseF (P), or not treated (NT) and then analyzed by western blot. Data are mean ± s.e.m. n.s., not significant, *$P \leq 0.05$, ****$P \leq 0.0001$ by ANOVA with Tukey's posthoc test $F = 50.85$

**I1061T NPC1 is degraded by ER-phagy**. The observation that the I1061T mutant was degraded in lysosomes without trafficking through the medial Golgi suggested that it might be degraded through autophagy. Recently, an autophagy-based ER degradation pathway called selective ER autophagy (ER-phagy) was shown to maintain ER homeostasis and degrade misfolded ER proteins in the lysosome[28,29]. To begin to assess whether NPC1 was degraded by autophagy, CTRL and I1061T fibroblasts were treated with the protein synthesis inhibitor cycloheximide and autophagy was induced by serum starvation. Serum starvation did not alter the clearance of WT NPC1 (Fig. 4a, Supplementary Fig. 3). However, serum starvation for 4 and 8 h significantly increased the clearance of I1061T protein; this clearance was inhibited by Baf (Fig. 4a, Supplementary Fig. 3). To determine whether I1061T trafficked to autophagosomes, confocal microscopy was used to visualize NPC1 protein. Following treatment with Baf, I1061T co-localized with the autophagosome marker LC3 (Fig. 4b), suggesting that I1061T protein trafficked through autophagy for degradation.

Although ER-phagy has been studied extensively in yeast[30–32], it is not as well characterized in mammalian cells. Recently, the reticulon protein family member FAM134B was shown to be a receptor for selective ER autophagy[29]. FAM134B is an ER membrane protein that directly binds LC3 to initiate autophagosome formation[28,29]. While no endogenous misfolded substrates have been characterized previously, FAM134B-mediated ER-phagy is hypothesized to be vital during cellular stress conditions[29].

To determine whether NPC1 is a substrate for ER-phagy, CTRL, I1061T, and P1007A/T1036M primary fibroblasts were treated with siRNAs targeting the macroautophagy regulator Beclin-1, the ER-phagy specific receptor FAM134B, or non-

targeted control. Target knockdown was confirmed by western blot (Fig. 5a, Supplementary Fig. 4a) or qPCR (Supplementary Fig. 4b). Knockdown of Beclin-1 or FAM134B significantly increased I1061T protein levels (Fig. 5a, Supplementary Fig. 4c), consistent with its degradation through ER-phagy. Similarly, knockdown of p97 caused a more modest increase in I1061T levels that trended in the same direction but failed to reach statistical significance (Fig. 5a). In contrast, neither WT nor P1007A/T1036M NPC1 was sensitive to knockdown of these proteins (Fig. 5a). While these experiments do not exclude the possibility that a small fraction of these proteins traffic through autophagy, the response of I1061T to these manipulations was much more robust.

Next, we performed cycloheximide chase experiments to determine whether FAM134B was required for I1061T degradation during serum starvation. Knockdown of FAM134B significantly slowed I1061T degradation 4 and 8 h after serum starvation (Fig. 5b). Notably, knockdown of FAM134B only slightly increased the accumulation of EndoH resistant species and did not rescue function as analyzed by filipin staining (Fig. 5c, d). Collectively, these data support the notion that misfolded I1061T species are degraded through ER-phagy.

As a complementary approach to assess the importance of ER-phagy to NPC1 degradation, FAM134B was over-expressed in CTRL and I1061T cells. This manipulation was previously shown to increase the flux of cargo through ER-phagy[29]. Over-expression of FAM134B effectively reduced WT and I1061T proteins as assessed by western blot (Fig. 6a) and I1061T protein as quantified by confocal microscopy (Fig. 6b). The observation that over-expression of FAM134B decreased WT NPC1 raised the possibility that a small fraction of this protein misfolds and is

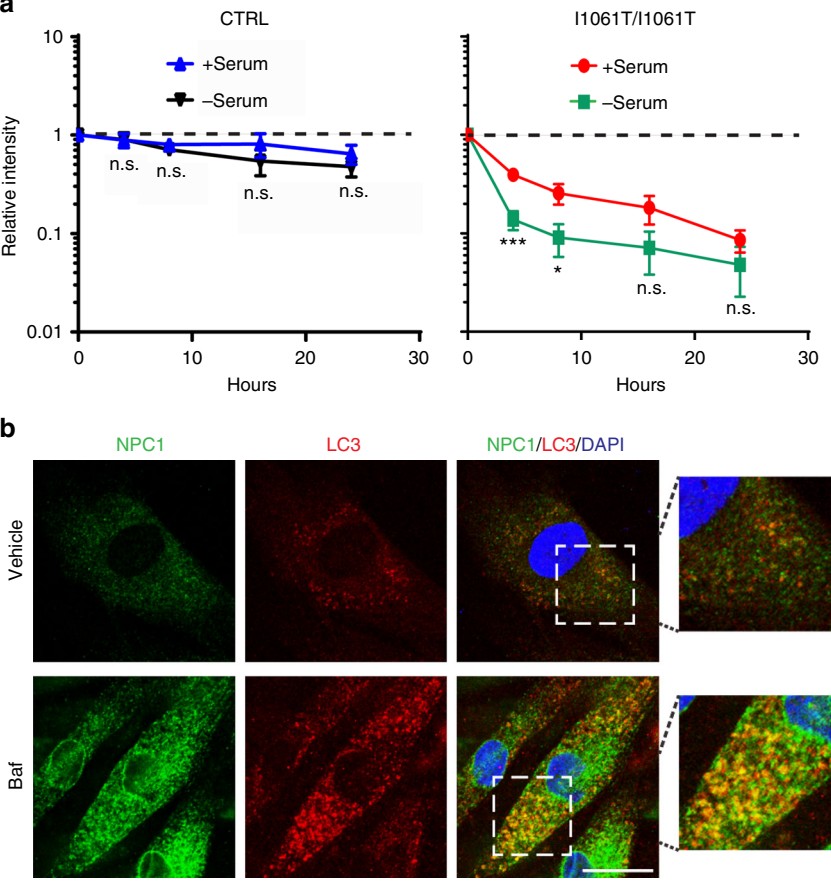

**Fig. 4** I1061T NPC1 traffics to autophagosomes. **a** CTRL or I1061T *NPC1* primary human fibroblasts were treated with cycloheximide in the absence or presence of serum. Lysates were collected at indicated times and analyzed by western blot. **b** I1061T *NPC1* fibroblasts were treated with vehicle or 100 nM bafilomycin A1 (Baf) for 24 h. Fixed cells were stained for LC3 (red), NPC1 (green), and DAPI (blue) then imaged by confocal microscopy. Co-localization is indicated by yellow color in the merged image. Scale bar = 25 μm. Data are mean ± s.e.m. from three independent experiments. n.s., not significant, *$P \le$ 0.05, ***$P \le$ 0.001 by ANOVA with Bonferroni posthoc test (I10) $F = 6.142$

trafficked through ER-phagy in a manner similar to I1061T. FAM134B utilizes an LC3 interacting region (LIR) to bind LC3 and initiate ER-phagy[29]. Over-expression of non-functional FAM134B lacking this LIR domain did not alter WT or I1061T NPC1 protein levels (Fig. 6c, Supplementary Fig. 5). We conclude that I1061T is an endogenous substrate for FAM134B-dependent ER-phagy.

**I1061T is degraded by ER-phagy in vivo**. To establish whether I1061T is degraded by ER-phagy in vivo, we used gene-targeted mice homozygous for the *Npc1 I1061T* allele. These mice display many pathological characteristics of Niemann–Pick type C disease including cholesterol storage and neurodegeneration[27]. Importantly, I1061T protein in liver and brain of these mice is reported to be EndoH sensitive, indicating that, as in patient fibroblasts, it is retained in the ER due to misfolding[27]. Similar to patient fibroblasts, treatment of cultured brain slices from WT and I1061T mice with Baf caused a robust accumulation of NPC1 protein (Supplementary Fig. 6). To visualize whether the I1061T protein is degraded by autophagy in vivo, we crossed *I1061T Npc1* mice to a transgenic line expressing the autophagy reporter GFP-LC3[33]. The resulting mice were treated with the microtubule inhibitor vinblastine for 2 h to slow autophagosome maturation[34–36]. This manipulation facilitated the ready identification of I1061T in autophagic vesicles of primary human fibroblasts

(Supplementary Fig. 7a). Similarly, both WT, and to a greater extent, I1061T NPC1 co-localized with GFP-LC3 in mouse livers (Fig. 7a, Supplementary Fig. 7b). Co-localization of WT protein likely reflected fusion of autophagosomes to NPC1-containing lysosomes. In contrast, the I1061T protein does not traffic to lysosomes, and therefore its robust co-localization with GFP-LC3 (Pearson correlation coefficient 0.48 ± 0.15) reflects the trafficking of mutant protein through autophagy for degradation.

To confirm the presence of I1061T in the autophagic pathway in the liver, we performed subcellular fractionation using differential centrifugation to separate cytosol (Cyt) and ER, and discontinuous density metrizamide gradients to enrich for autophagosomes (APG), autolysosome (AUT), and lysosomes (Lys). Fractionation was confirmed using markers of ER (calreticulin) and APG/AUT/Lys (LC3II, glucocerebrosidase). WT NPC1 was highly enriched in the Lys fraction (Fig. 7b, Supplementary Fig. 8a), whereas the I1061T protein was detected at high levels in the ER and was also readily detected in autophagic compartments (Fig. 7b, Supplementary Fig. 8a). Taken together, these studies indicated that I1061T traffics through autophagosomes in vivo, consistent with the notion that ER-phagy plays a significant role in its degradation. In fact, transmission electron microscopy of livers from control and I1061T mice revealed a significant increase in the fraction of autophagosomes containing ER as their cargo, along with the expected expansion of the autolysosomal compartment (Fig. 7c).

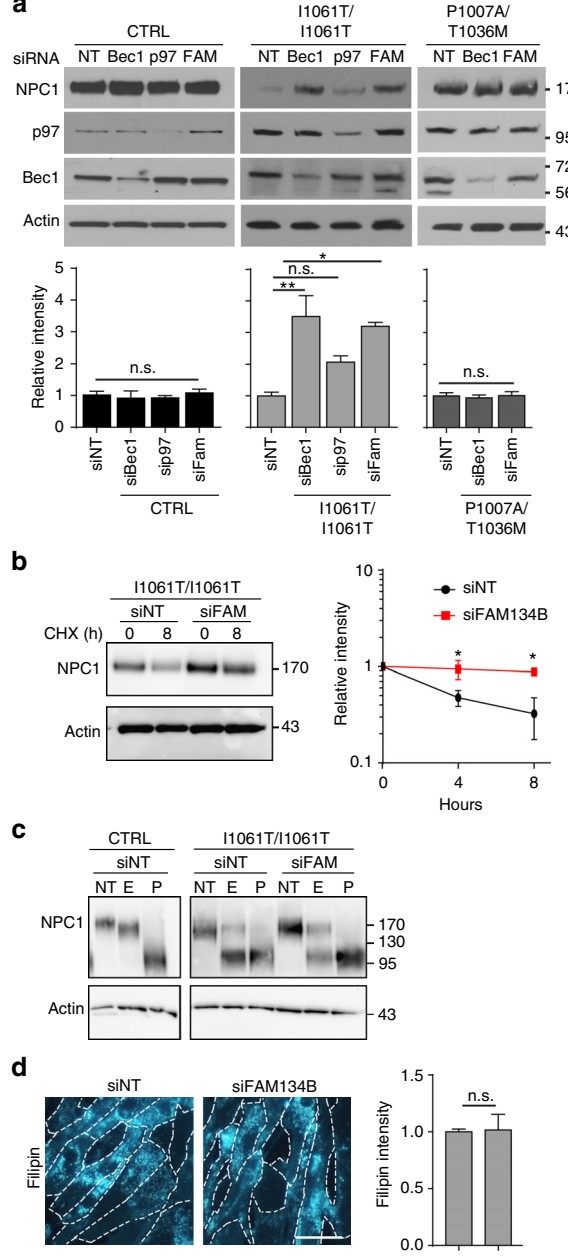

**Fig. 5** I1061T is degraded by ER-phagy. **a** CTRL, I1061T, or P1007A/T1036M *NPC1* fibroblasts were treated with the following siRNAs: non-targeting (NT), beclin-1 (Bec1), p97, and FAM134B (FAM) at $t = 0$ and $t = 24$ h. Lysates were collected at $t = 48$ h and NPC1 levels analyzed and quantified below. **b** I1061T *NPC1* fibroblasts were treated for 2 sequential days with non-targeting siRNA (siNT) or siRNA FAM134B (siFAM134B), then treated with cycloheximide and serum starved. NPC1 protein levels were analyzed by western blot, quantified at right. **c** Lysates from **a** were digested with endoglycosidase H (E), PNGaseF (P), or not treated (NT) and then analyzed by western blot. **d** I1061T *NPC1* fibroblasts were treated with siRNAs as in **a**. At $t = 48$ h, cells were stained with filipin. Intensity quantified at right. Scale bar = 50 μm. Data are mean ± s.e.m. from **a** 3–6; **b**, **c** 3 independent experiments. n.s., not significant, *$P \leq 0.05$. **$P \leq 0.01$ by **a**, **b** ANOVA with **a** Tukey or **b** Bonferroni posthoc test; **c** Student's $t$-test. *$P \leq 0.05$, **a** (I10) $F = 10.37$; **b** $F = 7.45$

This was in clear contrast with the control group, where mitochondria and bulk cytosolic contents were the most common autophagosome cargo. Relative to WT, I1061T mice also showed significant ER dilation (Fig. 7d). These ER and

autophagic features are consistent with increased ER-phagy in mutant mice.

**Niemann–Pick C mice and patients have altered ER-phagy.** Prior studies have shown that deletion of FAM134B disrupts neuronal ER-phagy and ER morphology causing degeneration of sensory neurons in mice[29]. However, it is unknown whether flux through this pathway is altered in neurodegenerative disease. We did not detect FAM134B protein in mouse liver, suggesting that other proteins may contribute to the regulation of ER-phagy in this tissue. In contrast, we readily detected FAM134B in mouse brain tissue (Fig. 8a). Subcellular fractionation (Supplementary Fig. 8) showed that FAM134B was highly enriched in the ER fraction from brains expressing WT *Npc1* (Fig. 8a, Supplementary Fig. 8a, b). In contrast, FAM134B was shifted from the ER to the APG, AUT, and Lys fractions in I1061T brains, consistent with an alteration in ER-phagy flux. This change in FAM134B fractionation was not indicative of altered gene expression (Fig. 8b), but instead may reflect increased flux and/or slowed substrate degradation, as demonstrated in other systems[37–42]. Finally, to determine if these changes were relevant in Niemann–Pick C patients, we analyzed cerebellar lysates from control (CTRL1, CTRL2) and Niemann–Pick C (NP1, NP2) patients for FAM134B, p62, and LC3-II. Western blot revealed that Niemann–Pick C patient cerebellum had increased levels of all three autophagic markers (Fig. 8c), suggesting that ER-phagy is altered in the diseased brain.

## Discussion
We demonstrate that I1061T NPC1 is degraded by two complementary pathways that target the misfolded, mutant protein to either the proteasome or autophagosome (Fig. 8d). As this mutant is encoded by the most common disease-causing allele in Niemann–Pick type C patients of western European ancestry[10], pathways regulating its proteostasis have emerged as an important focus of study. Bolstered by the finding that the I1061T mutant is functional but fails to properly traffic[12,13], we sought to understand the machinery controlling its degradation. Our analyses demonstrate that this multipass transmembrane glycoprotein misfolds in the ER and is exported for degradation by either MARCH6-dependent ERAD or FAM134B directed ER-phagy.

Our work builds on previous studies showing that the I1061T protein is degraded, in part, by the proteasome[11,12]. The ERAD machinery mediating this process had not been previously characterized. We found that knockdown or over-expression of the canonical ERAD components GP78, Derlin-1, or HRD1 does not alter I1061T protein levels (Fig. 2b, c). This is consistent with data showing that over-expression of GP78 fails to enhance degradation of exogenously expressed NPC1[11] and suggests the involvement of alternative ERAD components. Our finding that the E3 ubiquitin ligase MARCH6 regulates this process (Fig. 2d) is particularly intriguing. Prior studies have established a critical role for MARCH6 in ERAD of cholesterol regulatory proteins[19,21]. Its involvement in the degradation of the I1061T mutant was unanticipated and raises the possibility that this E3 may have an even broader role in degrading proteins that influence cholesterol homeostasis.

Most unexpected was the observation that the I1061T mutant is targeted for degradation by FAM134B-dependent ER-phagy. Previous studies demonstrated the importance of FAM134B in promoting selective macroautophagy by targeting perinuclear ER sheets to maintain homeostasis and neuronal function[29,43]. As modulating FAM134B expression by either siRNA knockdown (Fig. 5a) or over-expression (Fig. 6a) influenced I1061T NPC1 levels, we provide evidence that this is

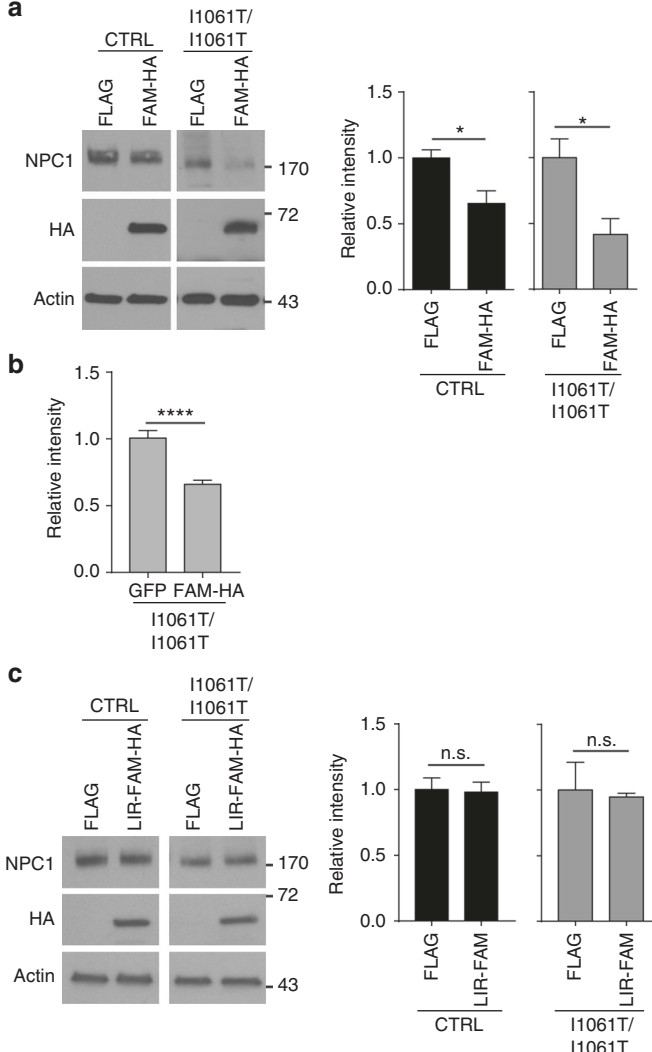

**Fig. 6** FAM134B over-expression enhances degradation of I1061T NPC1. **a**, **b** Over-expression of FLAG (control vector) or FAM134B-HA (FAM-HA) in CTRL and I1061T NPC1 fibroblasts. NPC1 protein levels were quantified by **a** western blot and **b** confocal microscopy. **c** Over-expression of FLAG (control vector) or FAM134B lacking the LC3 interacting domain (LIR-FAM-HA) in CTRL and I1061T NPC1 fibroblasts. NPC1 was analyzed by western blot. Quantified at right. Data are mean ± s.e.m. **a** N = 3–6 and **c** N = 3 independent experiments, **b** >32 cells per group. **a**–**c** Student's t-test; **a** (CTRL) t = 3.092, (I10) t = 3.113; **b** (I10) t = 5.511. n.s., not significant, *P ≤ 0.05, ****P ≤ 0.0001

a well-characterized, endogenous substrate for ER-phagy. We hypothesize that misfolded I1061T is trapped in ER sheets where it is recognized for degradation. Notably, a small quantity of WT NPC1 is also degraded through this pathway when FAM134B is over-expressed. This observation suggests the possibility that a fraction of WT NPC1 misfolds similarly to the I1061T mutant, triggering its degradation through ER-phagy. In support of the notion that ER-phagy is altered in NPC disease, we demonstrate that the fraction of autophagosomes containing ER as their cargo is significantly increased in livers from I1061T mice (Fig. 7c). In addition, fractionation studies of the I1061T mouse brain demonstrate a shift of FAM134B from the ER to fractions enriched for autophagosomes, autolysosomes, and lysosomes (Fig. 8a), consistent with the notion that flux through this pathway is higher in disease. Notably, while the number of autolysosomes is increased in

I1061T liver, we do not detect an accumulation of autophagosomes (Fig. 7c). This is consistent with prior studies in cell culture demonstrating largely intact autophagosome–lysosome fusion in NPC1 deficiency[37].

We did not readily detect FAM134B protein in the liver, raising the possibility that other receptors could be driving ER-phagy in this tissue. To initially explore this notion, we used the Human Protein Atlas[44] to analyze tissue expression of FAM134 family members FAM134B, FAM134A, FAM134C, as well as other recently described ER-phagy receptors SEC62[45], RTN3[46], and CCPG1[47]. Consistent with our work here, FAM134B is reportedly expressed at high levels in the brain relative to the liver. High brain expression is mirrored by SEC62, RTN3, and FAM134A. In contrast, both FAM134C and CCPG1 are highly expressed in liver relative to the brain. We speculate that misfolded I1061T NPC1 protein may interact with these alternative ER-phagy receptors in a cell or tissue-dependent manner. It is also interesting to note that the P1007A/T1036M mutant is resistant to effects of FAM134B knockdown (Fig. 5a). It is possible that this mutant misfolds in a different domain of the ER, such as peripheral tubules, and is subject to lysosomal degradation (based on sensitivity to Baf [Fig. 1d]) but controlled through alternative ER-phagy receptors. This would suggest non-overlapping functional roles of the set of ER-phagy receptors, a topic that warrants further investigation.

We note that additional proteins which misfold within the ER are degraded by macroautophagy[28,48,49], but whether this occurs through a FAM134B-dependent pathway remains unexplored. How MARCH6 or FAM134B select substrates for degradation is not understood. Thus, factors influencing triage of misfolded I1061T to either ER-phagy or MARCH6-dependent ERAD remain to be defined.

While knockdown of either FAM134B or MARCH6 slightly increased I1061T trafficking as determined by EndoH resistance (Fig. 5c, Supplementary Fig. 2c), they were not sufficient to correct I1061T function as measured by filipin staining (Fig. 5d, Supplementary Fig. 2d). These observations suggest that pharmacological interventions aimed at restoring NPC1 function in patients with this or similar mutations will likely need to target components upstream of these degradation pathways. Particularly appealing potential therapeutic targets include factors regulating the protein-folding environment within the ER, including calcium-dependent chaperones[13,50] such as calnexin or calreticulin. A number of small molecules aimed at these and other targets are currently under study in hopes of promoting proper folding of NPC1 missense mutants in order to restore function[14]. Patients harboring missense mutants that traffic inefficiently and retain partial activity, such as P1007A/T1036M, may also benefit from such therapeutics. It is our expectation that defining the pathways that regulate NPC1 proteostasis will guide these on-going drug discovery efforts and will continue to provide insights into fundamental mechanisms regulating ER protein quality control.

## Methods

**Reagents**. Bafilomycin A1 (B1793), MG132 (M7449), epoxomicin (E3652), vinblastine sulfate salt (cat. V1377), chloroquine diphosphate (C6628), eeyarestatin I (324521), 2-hydroxypropyl-β-cyclodextrin (H-107), (Z)-4-hydroxytamoxifen (H7904), cycloheximide (C7698), chloroquine (C6628), and filipin (F9765) were from Sigma. EndoH (P0702) and PNGaseF (P0704) were from New England Biolabs. Brefeldin A (20350) was from Cayman Chemical.

**Antibodies**. *Primary antibodies*: The following primary antibodies were used (antigen, dilution, vendor): Sel1L, 1:1000, Abcam ab78298; Actin, 1:4000, Sigma A5441; NPC1, 1:500 western, Abcam ab36983 (discontinued); NPC1, 1:200 imaging, Abcam ab134113; Flag, 1:1000, Sigma F1804; GP78, 1:1000, Cell Signaling 9590; Der-1, 1:1000, Cell Signaling 8897; HRD1, 1:1000, Cell Signaling. D302a; HA, 1:1000, Biolegend 901501;

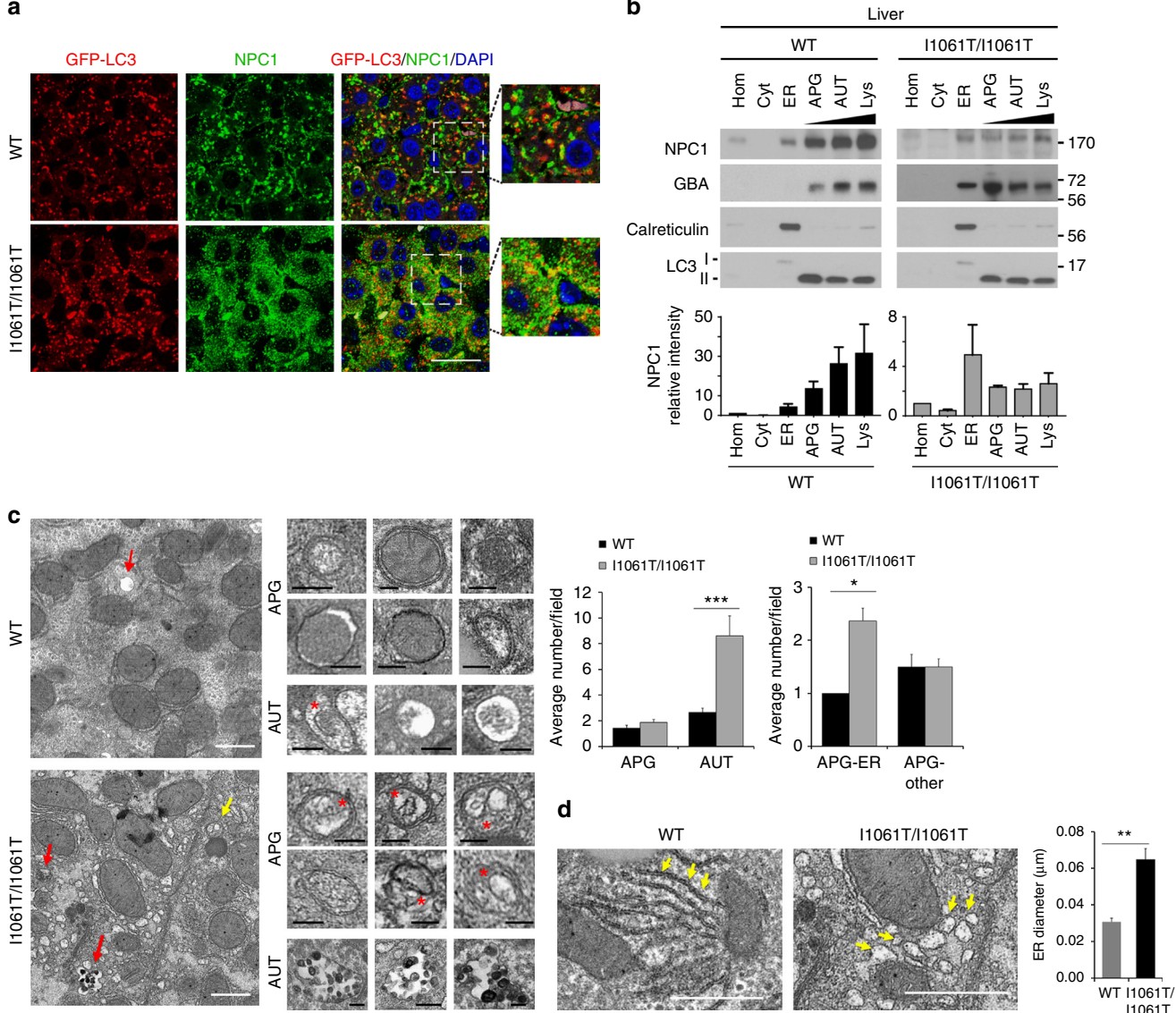

**Fig. 7** I1061T NPC1 traffics to autophagosomes in vivo. **a** Seven-week-old I1061T/I1061T-*Npc1*, GFP-LC3 or wildtype-*Npc1* (WT), GFP-LC3 mice were treated with vinblastine for 2 h. NPC1 (green) and LC3-GFP (red) in hepatocytes were visualized by confocal microscopy. Nuclei were stained with DAPI (blue). Due to expression differences and to prevent overexposure, WT and I1061T were imaged at different exposures. Scale bar = 25 μm. **b** Whole liver homogenates (Hom) from 7-week-old WT or I1061T-*Npc1* mice, or fractions enriched for cytosol (Cyt), ER, autophagosomes (APG), autolysosomes (AUT), or lysosomes (Lys) were analyzed by western blot. Blots were probed for LC3 (autophagic compartments), calreticulin (ER), or glucocerebrosidase (GBA; lysosomes). NPC1 band intensity was normalized to total protein as determined by Ponceau S stain (see Supplementary Fig. 8a). **c** Left: Electron micrographs of livers from WT and I1061T-*Npc1* mice. Yellow arrows: autophagosomes (APG); red arrows: autolysosomes (AUT). Inserts: Representative images of APG and AUT. * indicates ER inside APG. Graphs: Average number of APG and AUT per field (left) and average number of autophagosomes containing ER or other cargo (right) were calculated by morphometric analysis of N = 38 fields from 3 mice. Scale bar = 1 μm. **d** Electron micrographs of livers from WT and I1061T-*Npc1* mice. Arrows indicate ER. Right: Average ER diameter calculated by morphometric analysis of N = 38 fields from 3 mice. Scale bar = 1 μm. Data are mean ± s.e.m. Student's t-test (**c**); average AUT/field t = 3.2, average APG-ER t = 1.8; ER diameter t = 6. *P < 0.01, ***P < 0.0001. Scale bar: white—1 μm, black—0.2 μm

LC3, 1:1000 western, Novus NB600; LC3, 1:200 imaging, Gentex GTX17380; Beclin-1, 1:1000, Santa Cruz H-300; P97, 1:1000, Cell Signaling 2648; Ubiquitin, 1:500, Dako (discontinued); GAPDH, 1:5000, Novus NB600; Vinculin, 1:2000, Sigma V9131; FAM134B, 1:500, Abcam ab151755; FAM134B, 1:500, Proteintech 21537; p62, 1:1000, Sigma P0067; GBA, 1:1000, Sigma G4171; Calreticulin, 1:1000, Cell Signaling Technology 2891, GFP, 1:1000, Abcam ab13970. NPC1, 1:500, Abcam ab134113 was used for Figs. 3b and 5c, Supplementary Fig. S1b, c.

Secondary antibodies: The following secondary antibodies were used (antibody, dilution, vendor): Alexa Fluor 488 goat anti-rabbit IgG (H+L), 1:500, Invitrogen A11008; Alexa Fluor 594 Fab'2 fragment of goat anti-mouse IgG (H+L), 1:500, Invitrogen A11020; Alexa Fluor 594 goat anti-chicken IgY (H+L), 1:1000, Invitrogen A11042; Goat anti-mouse IgG (H+L)-HRP conjugate, 1:2000, Bio-Rad 170-6516; Goat anti-rabbit IgG (H+L)-HRP conjugate, 1:2000, Bio-Rad 170-6515.

**Plasmids.** FAM134B-HA (FAM-HA)[29] was from Ivan Dikic (Fachbereich Medizin Frankfurt Goethe Universität). FAM134B-ΔLIR-HA (FAM-ΔLIR-HA) was purchased from GenScript. GP78-Flag, Der-1-HA, HRD1-Flag were from Billy Tsai (University of Michigan).

**RT-qPCR.** RNA was collected using TRIzol® (Thermo Fisher) and converted to cDNA using the High Capacity Reverse Transcription kit (Applied Biosystems 4368814). Quantitative real-time PCR was conducted in technical triplicates using 25 ng cDNA, TaqMan™ probes (Thermo Fisher) for human and mouse FAM134B (Hs4351372 and Mm00481400), GAPDH (4325792), 18s (4310893), MARCH6 (Applied Biosystems Hs01020084). RT-qPCR was performed using an ABI

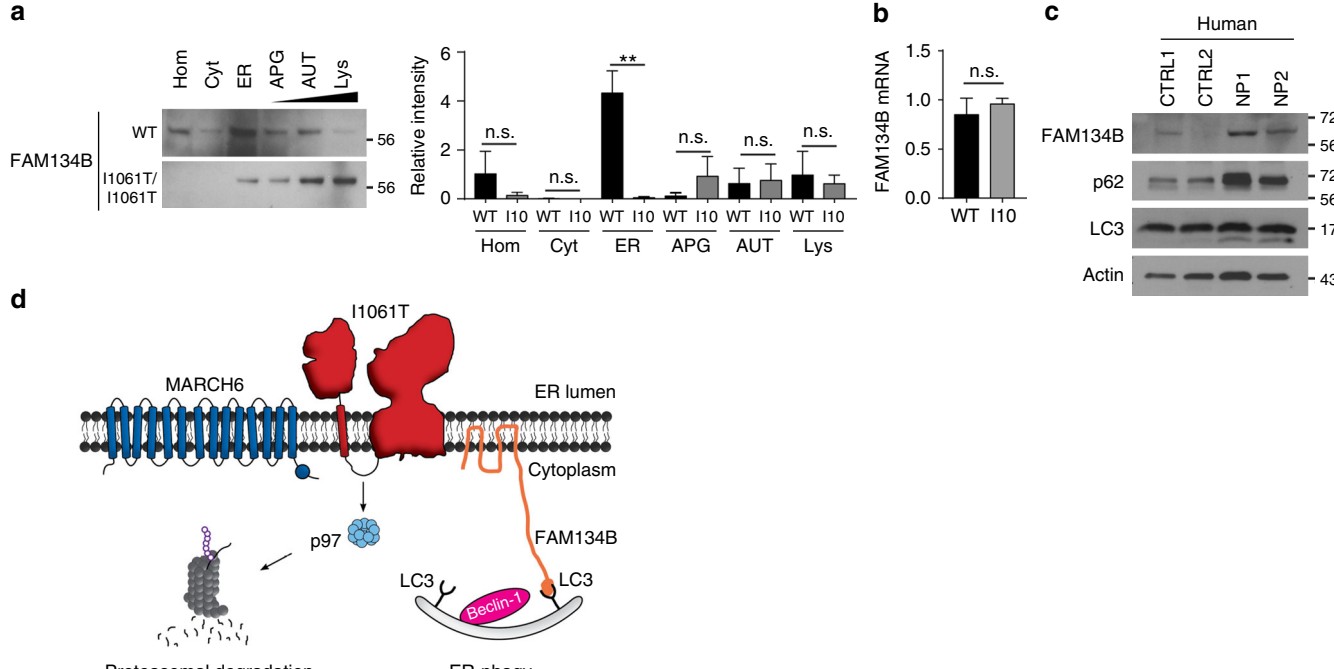

**Fig. 8** Flux through ER-phagy is altered in Niemann–Pick C mouse and patient brain. **a** Whole brain homogenates (Hom) from 7-week-old WT and I1061T-*Npc1* (I10) mice, or fractions enriched for cytosol (Cyt), ER, autophagosomes (APG), autolysosomes (AUT), or lysosomes (Lys) were analyzed by western blot. Blots were probed for FAM134B. Quantified at right relative to total protein as determined by Ponceau S stain (see Supplementary Fig. 8a, b). **b** Relative FAM134B mRNA levels in the brain of 7-week-old WT and I1061T-*Npc1* mice. **c** Cerebellar lysates from control (CTRL1, CTRL2) and Niemann–Pick C (NP1, NP2) subjects were analyzed by western blot. **d** Model of I1061T degradation by ERAD and ER-phagy. Data are mean ± s.e.m. from **a**, **b** N = 3 independent experiments. **a** ANOVA with Tukey's post-hoc, F = 3.85, **b** Student's t-test, n.s. = not significant, **P ≤ 0.01

7900HT Sequence Detection System and relative expression calculated by the 2^(−delta delta Ct) method using SDS software.

**Endoglycosidase H assay**. The endoglycosidase H assay was modified from Yu et al.[13]. Briefly, three reactions (negative control, EndoH (New England BioLabs P0702L), PNGaseF (New England BioLabs P0704L)) containing 40 μg cell lysate, 2 μl glycoprotein denaturing buffer, and water were combined to create a 20 μl reaction and incubated at 37 °C for 15 min. After incubation, the EndoH samples received 4 μl 10× G3 reaction buffer, 10 μl water, and 6 μl EndoH to make a total reaction volume of 40 μl. In the negative control reactions, the EndoH was replaced with water. PNGaseF reactions received 4 μl of 10× G2 reaction buffer, 4 μl 10% NP40, 8 μl water, and 4 μl PNGaseF. All reactions were placed at 37 °C for 3 h. Reactions were terminated with 15 μl 4× Laemmli sample buffer (Bio-Rad 161-0747) containing DTT before storage at −80 °C.

**Mice**. *Npc1*-I1061T mice[27] were a gift from Daniel Ory (Washington University in St. Louis) and backcrossed to C57BL/6 (≥10 generations). GFP-LC3 mice[33] were on the C57BL6 background. All procedures involving mice were approved by the University of Michigan Committee on Use and Care of Animals (PRO00006114) and conducted in accordance with institutional and federal guidelines.

**Vinblastine treatment**. Cells were incubated with 50 μM vinblastine for 2 h, washed 3× in HBSS, and fixed in 4% PFA. Mice were injected with vinblastine (0.05 mg/g) I.P. at 7 weeks of age. The brain and liver were collected 2 h post-injection from non-perfused mice. The tissues were quickly rinsed in room temperature DMEM with 4.5 g/l D-glucose and L-glutamine (Gibco 11965-092) and transferred to 15 ml conical tubes filled with DMEM, and kept at room temperature until fractionation.

**Subcellular fractionation**. Fractions enriched in autophagic vacuoles were isolated from mouse brain and livers[51]. Briefly, samples were subjected to centrifugation to separate a fraction enriched in autophagic vacuoles, lysosomes, and mitochondria. After centrifugation in a discontinuous metrizamide density gradient, autophagosomes were recovered in the 20–15% metrizamide interface, autolysosomes in the 24–20%, and lysosomes in the 26–24%. The purity of the lysosomal fractions was assessed by measuring total and specific enzymatic activities of the lysosomal enzyme β-hexosaminidase. The integrity of the lysosomal membrane was determined by measurement of leakage of this lysosomal enzyme in the medium[52].

Cytosolic fractions were obtained by centrifugation for 1 h at 100,000×g of the supernatant obtained after separating the mitochondria–lysosome-enriched fraction, which allows for the recovery of the cytosol in the supernatant and ER in the pellet.

**Transmission electron microscopy and morphometric analysis**. Vinblastine treated mice (as above) were anesthetized and perfused with sterile saline. Livers were collected, sectioned (0.5 mm × 0.5 mm × 2.0 mm), and immersed in a 2.5% glutaraldehyde (dissolved in 0.1 M phosphate buffer pH 7.4) for 2 h. Tissue was incubated in Karnovski's fixative (2.5% glutaraldehyde + 3% PFA) for at least 1 h at room temperature, then overnight at 4 °C. Samples were washed with 20× volume Sorenson's buffer 3 times before post-fixing in 2% osmium tetroxide in Sorenson's buffer for 1 h. Tissue was again washed 3× with 20× volume Sorenson's buffer, then dehydrated through ascending concentrations of ethanol, treated with propylene oxide, and embedded in EMbed 812. Semi-thin sections were stained with toluidine blue. Regions were selected for ultra-thin sections (70 nm) and post-stained with uranyl acetate and Reynold's lead citrate. Tissues were examined using a JEOL JEM-1400 Plus transmission electron microscope at 80 kV.

For quantification, images were taken from 3 mice per group at 3000× (quantify autophagy and structural differences) and 6000× (define autophagic cargo). Autophagic vacuoles (vesicles, 0.5 μm) were classified as autophagosomes when they met two or more of the following criteria: double membranes (complete or at least partially visible), absence of ribosomes attached to the cytosolic side of the membrane, luminal density similar to cytosol, and identifiable organelles or regions of organelles in their lumen[53]. Vesicles of similar size but with a single membrane (or less than 40% of the membrane visible as double), luminal density lower than the surrounding cytosol, or multiple single membrane-limited vesicles containing light or dense amorphous material were classified as autolysosomes. Vesicle area and ER diameter were measured with the freehand selection tool of ImageJ. Morphometric analyses were performed on blinded samples.

**Cells**. The following cell lines were obtained from the NIGMS Human Genetic Cell Repository at the Coriell Institute for Medical Research: GM08399 (CTRL), GM18453 (I1061T/I1061T), GM17912 (P1007A/T1036M), GM03123 (C.1947+5G>C/I1061T[54]), GM17926 (I1061T/Y509S), and GM17924 (451ΔAG/Y825C). Fibroblasts were cultured in MEM (Gibco 10370), PSG (Gibco), and 20% premium FBS (Atlanta Biologicals). Cells were not cultured past passage 25.

Previously characterized Sel1L conditional null, immortalized mouse embryonic fibroblasts[17] were treated with 200 nM tamoxifen for 48 h to delete the floxed allele.

Cycloheximide chase: Cells were treated with 60 μg/ml cycloheximide for the indicated times. Serum starvation was induced by replacing cell culture media with cycloheximide containing MEM media without FBS.

**Patient tissues.** Human tissue was obtained from the NIH Neurobiobank at the University of Maryland, Baltimore, MD. The NIH Neurobiobank at the University of Maryland is overseen by Institutional Review Boards at the University of Maryland (HM-HP-00042077) and the Maryland Department of Health and Mental Hygiene (5-58), which ensure that informed consent was obtained from patients who donate tissue. Samples were frozen cerebellum from non-affected (CTRL) and Niemann–Pick C (NP) patients at 11 and 19 years of age (YOA). CTRL1: UMB patient #1793 (11 YOA); CTRL2: UMB patient # 5470 (19 YOA); NP1: UMB patient # 4237 (19 YOA); NP2: UMB patient # 5372 (11 YOA).

**Western blot.** Cell culture media was aspirated and replaced with PBS. Cells were removed with a cell scraper and centrifuged at $1000 \times g$ for 5 min at 4 °C. The cell pellet was resuspended in 70 μl RIPA (Teknova) with complete protease inhibitor (Thermo Scientific 11836153001), and 0.625 mg/ml N-ethylmaleimide (Sigma E3876) and sonicated. Protein concentrations were determined by DC™-protein assay (Bio-Rad) and normalized. To visualize I1061T-NPC1, 50 μg was loaded per well. Proteins were separated on 4–20% gradient SDS page gels (Bio-Rad) and transferred to Immobilon®-P 0.45 μm PVDF (Merck Millipore). Immunoreactivity was detected with ECL (Thermo Scientific) or TMA-6 (Lumigen) with X-ray film or an iBright (Thermo Fisher Scientific). Quantification was performed using ImageJ after subtracting background. Band intensity was normalized to the indicated loading control. In some cases, multiple exposures are presented in the Supplement. If brightness and contrast were modified, adjustments were performed equally to the entire image and controls. Uncropped blots are shown in the Supplementary Information.

**Transfection.** *Plasmids*: Cells were grown to 70% confluency in 6-well plates and media was replaced with 1.4 ml cell culture media without pen/strep. Cells and media are equilibrated for 1 h before transfection. GeneIn™ (Amsbio, discontinued) was used to transfect primary fibroblasts. Briefly, 1.75 μg of endotoxin-free plasmid, 8 μl red reagent, and 200 μl OPTI-MEM® (Gibco) were vortexed for 2 s. After 5 min incubation at room temperature, 4 μl of blue reagent was added and the reaction was briefly vortexed. Following 15 min incubation, 200 μl of reaction mix was added dropwise to 1.4 ml cell culture media ($t = 0$). At $t = 24$ h, the transfection protocol was repeated and lysates were collected at $t = 48$ h.

siRNA: siRNAs were predesigned ON-TARGETplus SMARTpool-Human containing 4 individual siRNA sequences (Dharmacon Non-targeting SMARTpool D-001810-10-05, FAM134B L-016936-02, P97 L-008727-00, Beclin-1 L-010552-00, HRD1 L-007090-00, GP78 L-006522-00, Derl-1 L-010733-02, MARCH6 L-006925-00). The individual siRNAs in the FAM134B SMARTpool were tested in Supplementary Fig. 4b (Product number and targeting sequence: 1# J-016936-18, AGUUGUAGACUUAGGCUUA; #2 J-016936-19, UCAGAAGAAACGUGA GAGA; #3 J-016936-20, GAGAGUGAAUUGGGACUUA; #4 J-016936-21, CCGAAAUUAUCAAGGUAUC). siRNAs were transfected using TransIT-X2® (Mirus) into 3.5 cm wells as follows. Cells were plated to 70% confluency and at $t = 0$ h media was replaced with 1.5 ml complete cell culture media. 9 μl TransIT-X2 was added to 250 μl OPTI-MEM and 8 μl of 20 μM siRNA solution to create the reaction solution. The reaction solution was gently inverted 3× and incubated at room temperature for 25 min. 250 μl reaction solution was added dropwise to one 3.5 cm well. This process was repeated at $t = 24$ h and cell lysates were collected at $t = 48$ h.

**Immunofluorescence staining.** Cells were washed 3× with HBSS and immediately fixed with cold 100% methanol for 20 min at −20 °C. Cells were briefly washed 3× with PBS and put in 2.5 mg/ml glycine for 10 min at room temperature. After 3× washes with PBS, cells were incubated with blocking solution (0.02% saponin, 5% NGS, 1% BSA) for 1 h. Cells were incubated with primary antibodies diluted in blocking solution overnight at 4 °C. Slides were washed 3× for 10 min with PBS + 0.02% saponin and incubated with secondary antibody diluted in blocking solution for 1 h at room temperature. After 2 washes with PBS + 0.02% saponin and 2 washes with PBS, slides were mounted with Vectashield + DAPI (Vector Laboratories).

For tissue preparation, mice were perfused with 4% PFA, and tissue was placed in 4% PFA overnight at 4 °C prior to processing. Paraffin-embedded tissues were cut on a Reichert-Jung 2030 microtome into 5 μm sections and placed on Fisher Scientific Superfrost Plus microscope slides. Sections were adhered onto slides in an oven at 55–60 °C for 1 h. Samples were deparaffinized using organic solvents. Antigen retrieval was performed by boiling in 10 mM sodium citrate (pH 6.0) for 10 min, then incubating the samples in hot citrate solution for an additional 20 min. The slides were washed 3× for 2 min in deionized water before blocking. For staining, sections were incubated in 10% goat serum, 1% BSA, and 0.1% Triton for 10 min. This solution was replaced by block solution without detergent and sections were incubated for an additional 50 min at room temperature. Slides were

incubated overnight at 4 °C in primary antibody diluted in block solution. After three 5 min washes in PBS, slides were incubated for 1 h at room temperature with a fluorescent secondary antibody diluted in block solution with 5% goat serum. Slides were subsequently washed in PBS three times, mounted with ProLong Gold (Thermo Fisher), and sealed[55].

**Filipin staining.** Fibroblasts were grown in Nunc™ Lab-Tek™ II 4 well chamber slides™ (Thermo Fisher). Before staining, cells were incubated with 5 μg/ml wheat germ agglutin® 488 conjugate for 10 min at room temperature, then immediately fixed with 4% PFA for 20 min. After three PBS washes, residual PFA was quenched with 1.5 mg/ml glycine for 10 min. Next, chambers were incubated with 1 ml filipin staining solution (5% FBS + 40 μl filipin solution (1 mg filipin + 40 μl DMSO in PBS)) for 2 h at room temperature[55]. After three washes in PBS, the chamber was removed and slides were mounted with Pro-Long® Gold (Thermo Fisher).

**Microscopy.** Confocal images were collected using a Nikon A-1 confocal with diode-based laser system. Co-localization and intensity were determined using Nikon A1 elements software. Filipin staining was visualized using an epi-fluorescence Zeiss Axio Imager Z1 microscope with an automated stage[37]. To circumvent photobleaching, slides were focused using wheat germ agglutinin (Molecular Probes) under the green filter. Sequential tiling was used to capture 4 images in the UV channel at 20× magnification. Each experiment consisted of 16 images (each with ≥90% confluence; ~133–182 cells/image), which were quantified and averaged using NIH ImageJ[13]. Data reported are from three independent experiments. Liver co-localization was determined using the co-localization module in CellProfiler Analyst software. Brightness and contrast were applied equally across the entire image and to controls using Photoshop.

**Slice culture.** Sagittal brain slice cultures (300 μm thickness) were prepared and incubated in media containing either vehicle or bafilomycin (5 mM) at 37 °C in 95% $O_2$/5% $CO_2$[56]. Slices were incubated for 48 h, after which two brain slices per condition were transferred to a tube containing 300 μl RIPA and 0.6 g of 1.6 mm stainless steel beads (Next Advance). Samples were placed in a bullet blender (Next Advance) at a speed setting of 3 for 3 min. Steel beads were removed with a magnetic wand (Next Advance) and sonicated. Cellular debris was removed by a 5 min, $1000 \times g$ spin. Normalized protein was resolved by SDS-PAGE and analyzed by western blot.

**Cell survival.** The XTT assay (ATCC) was used to assess cell survival. 50 μl of XTT was added to 100 μl of cell culture media for 6 h in a $CO_2$ incubator at 37 °C. Plates were read on a Synergy HTX multimode plate reader (BioTek) at 490 and 650 nm.

**Statistics.** GraphPad Prism 7.0 was used to determine significance ($P < 0.05$), $F$ (F-statistic), and $t$ (T-statistic) values. Unpaired Student's $t$-test, one-way, or two-way ANOVA with Tukey or Bonferroni posthoc analysis were used as indicated in the figure legends. All error bars are s.e.m.

## Data availability

The data supporting the findings of this study are available from the corresponding author upon reasonable request.

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

## Acknowledgements

We thank Dr. Daniel Ory for the gift of I1061T *Npc1* mice, Drs. Billy Tsai, Andrew Brown, and Ivan Dikic for providing plasmids, and Dr. Cristin Davidson for sharing the NPC1 staining protocol. We are grateful to Kayla Capper for creating the scientific illustration. This work was supported by the U.S. National Institutes of Health (R01 NS063967 to A.P.L., T32 NS007222 to M.L.S., R01 GM113188 to L.Q., R01 NS085054 to V.G.S., and P01 AG031782 to A.M.C.), SOAR-VPN (to A.P.L. and A.M.C), the University of Michigan Protein Folding Diseases Initiative, and the University of Pennsylvania Orphan Disease Center (to A.P.L.).

## Author contributions

M.L.S. designed and performed experiments, analyzed data, and wrote the manuscript. K.L.K., D.D., R.C., S.K., V.G.S., and A.M.C. performed experiments and edited manuscript. L.Q. provided a unique reagent and edited the manuscript. A.P.L. designed experiments, analyzed data, and edited the manuscript.

## Additional information

**Competing interests:** The authors declare no competing interests.

