## [Peer Review File · Nature Communications]

Reviewers' comments:

Reviewer #1 (Remarks to the Author):

NCOMMS-17-16374-T
Schultz et al.

This manuscript examines the cellular machinery involved in the processing of the disease allele (NPC1 I1061T) most commonly present in Niemann-Pick C1 disease, a neurodegenerative disorder. Previous work in the field demonstrated that the mutant protein is misfolded and degraded by the ERAD. In the present study the authors extend this work by showing that the MARCH6 E3 ubiquitin ligase is responsible for degradation of the I1061T protein. They show that inhibition of lysosome acidification and that inhibition of p97 also increases I1061T protein levels, suggesting that the protein may also be degraded by autophagy. They provide evidence that I1061T protein is degraded by ER-phagy and identify NPC1 I1061T as the first endogenous substrate for this FAM134B-dependent process. The authors show that in the brains of the NPC1 I1061T knockin mouse, FAM134B shifts to the autophagosome, autolysosome, lysosome fractions, and in the cerebella of human NPC1 subjects markers for these compartments, including FAM134B, are elevated.

This study is an important conceptual advance in our understanding of Niemann-Pick C1 disease, specifically the handling of the misfolded I1061T protein in the ER. The *in vivo* studies in the mouse I1061T model and the human tissue data are consistent with an ER-phagy mechanism and provide strong support for the cell-based observations. Regulation of protein folding in the ER has emerged as a potential target for therapeutic intervention in NPC1 disease because the vast majority of disease is caused by missense mutations, and the majority of these are believed to be misfolding mutants. The discovery of the I1061T protein as a substrate for FAM134B provides new insights into the handling of this mutant in the ER and could represent a point for intervention. NPC1 I1061T protein is apparently the first reported endogenous substrate for FAM134B, but it seems likely that there are other endogenous substrates, which could have implications beyond NPC1 disease. An important question is whether this pathway is operative only for the misfolded mutant NPC1 protein, or whether WT protein is also a substrate. The authors present data suggesting only the mutant protein is being handled by FAM134B, but the data is not convincing (see point 1 below). Previous work (ref 12) showed that ~50% of newly synthesized WT protein does not leave the ER and is degraded. This suggests that the energetics of NPC1 protein folding may involve states that are similar to that of the I1061 protein and thus these states are recognized as misfolded and targeted for degradation. A deeper examination of this mechanism could address this question.

Specific points:

1. An important question is whether the ER-phagy pathway handles both normal and mutant substrates. In response to knockdown of FAM134b, WT NPC1 protein band (fig 4a) clearly is increased compared to untreated (the actin band slightly reduced further yielding a higher ratio), yet the quantification indicates no change. It may be the case that the blot

shown is not representative; however, the complicating factor here that the western is measuring total NPC1 protein, the vast majority of which is in the lysosome, rather than informing with respect to nascent protein. Metabolic labeling coupled with endoglycosidase H treatment studies could definitively answer whether FAM134b knockdown affect degradation of the newly synthesized, ER-associated WT protein. This would be a nice addition to this study.

2. It seems likely that there are endogenous substrates for the ER-phagy pathway. An excellent candidate would be CFTR, another polytopic transmembrane protein for which the WT protein is largely degraded in the ER. Did the authors examine whether CFTR might also be a substrate for FAM134B?

3. In addition to I1061T, four compound heterozygote cell lines were also studied, but no rationale is provided for the selection of these mutants. With the exception of the C. 1947+5G>C/I1061T line, which only expresses I1061T protein and does not add to the data obtained from the homozygous I1061T cells, the other three express two mutant species, confounding interpretation of the western data. Human cell lines expressing a single NPC1 mutant are commercially available, albeit limited (For example, P1007A compound heterozygote with a null allele is available at Coriell). Alternatively, if the goal were to examine a panel of mutants, the author could consider transient expression in stable lines (Pipalia JLR 2017), though it is not clear that this information is critical for the manuscript. As is, the data seems of limited value and could be removed.

4. Consideration should be given to testing the FAM134B dependence for the NPC1 protein in vivo. This could be achieved by ASO treatment of I1061T/GFP-LC3 mice and monitoring the NPC1 localization in liver tissue. These experiments would also parse out whether the NPC1 protein expressed in the liver and shown to co-localize with the GFP-LC3 marker is due to fusion of autophagosomes to NPC1-containing lysosomes, as suggested by the authors, or a result of FAM134B-dependent trafficking. These studies would be of particular interest if metabolic labeling studies suggest that WT NPC1 protein is a substrate.

Minor points:

1. In several figures, the representative NPC1 bands shown in the westerns do not agree with the quantification. This seems to be the case in fig 4a (see point #1 above) and fig 2d for the siMARCH6-treated normal NPC1 protein. As noted above, the issue may be the background created by the vast excess of lysosome-associated normal protein, swamping out a signal for increased ER-associated protein.

2. In WT cells, bafilomycin results in a robust increase in higher MW bands (Fig 1b) and EndoH-resistant NPC1 (Fig 3b), which are not addressed. One interpretation is that inhibition of lysosome acidification is interfering with turnover of fully-glycosylated NPC1 in this compartment. This should be discussed.

3. In Fig 6d, what criteria were used to determine that the ER components were contained within autophagosomes?

Reviewer #2 (Remarks to the Author):

In the submitted manuscript "coordinate regulation of I1061T NPC1 degradation by selective ER autophagy and MARCH6-dependent ERAD" the authors analyse the fate of different disease-related NPC1 mutants, with the main focus on I1061T NPC1. The authors showed proteasomal (ERAD) as well as autophagosomal (ER-phagy) participation in the turnover of WT and mutant and analysed the composition of the respective machineries. The manuscript contains novel mechanistic findings behind the Niemann-Pick type C disease and indicates potential therapeutic targets.

While specifically the involvement of ER-phagy in the turnover of disease-related NPC1 mutant proteins is a surprising and therefore important finding for the field, the manuscript in its current form is lacking the maturity and in some parts quality for publication in nature communication.

At this point it is important to mention that the authors completely avoided mentioning the contradiction of their finding of autophagosomal degradation of NPC1 to current literature. This is specifically irritating, as the main authors have published a very nice review on Niemann-Pick type C disease about a year ago, listing publications reporting that autophagy is not involved in NPC1 degradation. In contrast, Schultz et al found autophagy to be the major degradation pathway (80%) of NPC1 mutant I1061T in their setting.

While there are numerous reasons and explanations why this might be the case, it has to be discussed and some key experiments of former studies (usage of the same compound to block autophagy, namely chloroquine and NH₄Cl) have to be repeated in the settings used in this study.

For a reported half-life of 9h for major part of WT and 6.5h for I1061T mutant protein, the authors choose a relatively long treatment time of 24h to detect differences in NPC1 WT7mutant protein levels. This choice and its impact on the experimental outcome (e.g. analysis of different protein pools in comparison to other studies) should be explained to readers.

A control experiment for fitness of cells (e.g. WB of presented lysates for markers of different cell death pathways + respective positive controls) should be included as a supplementary figure to exclude potential artifacts.

Further major and minor points:

Figure 1

Major point:

To understand the contribution of the two degradation pathways and potential differences for individual protein pools, this time course should be done as well in the presence of MG132 alone and upon double treatment with MG132+Baf.

Minor point:

It would be also very informative to have a similar (potentially shorter) time course in the presence of CHX to indicate the contribution of protein synthesis and degradation.

Figure 2

Major point:

Panel a-c) As knockdown of MARCH6 (Fig 2d) has no effect on WT levels, while the WT protein is stabilized upon MG132 treatment (Fig 1a), the same experiments should be done with WT cells.

Minor point:

Current panel b+c) Since the main focus in this figure lies on the mutant protein, CTRL+FLAG should be excluded from the bar graph (while staying in the WB panel) and intensities should be normalized to I1061T + FLAG / +NT.

Figure 3

Major point:

Panel a) Relative intensity should be normalized to I1061T + Vehicle= high to low, less error prone

Minor point:

Panel b) Presentation of data of individual cell lines would benefit from being on the same gel and thereby presented with same exposure time. As it looks now, e.g. I1061T +MG shows lower NPC1 signal than when treated with Vehicle = contradiction to Fig.1.

Figure 4

Major point:

The whole figure, in specifically quantification of data should be carefully reviewed.

Panel a) In case presented blots are indeed representatives for the performed experiments, one has to doubt the quantification below. By eye and based on a quick ImageJ quantification the levels of WT NPC1 upon silencing of Bec/p97/Fam change 1.17x/1.25x/1.6x and even more profound when taking the loading control into account (1.34x/1.4x/1.9x).

These numbers would actually fit to the data presented in Fig1: WT protein levels are increasing upon Baf treatment 2.5x.

Panel b) Inconsistent NPC1 Levels. I10-Fam/I10-NT is 6.5x in panel a) and less than 1x in panel b)

Panel c) please specify in fig legend appx. number of cells / number of fields quantified for each of the three independent experiments.

Figure 5

Minor point/suggestion:

Even though not significant, WT NPC1 levels seem to be slightly affected by FAM134B

overexpression. Specifically when comparing it to the quantification of LIR mutant. This would fit to data presented in Fig1a and WB panels of Fig 4a and could be noted in the text.

Further minor points:

- For clarity CTRL should be replaced by WT in figures and text whenever applicable.
- The authors come to the conclusion that accumulated mutant protein is non-functional (Fig. 3a+b). This is in contrast to published data and could be an interesting point for the discussion.

Reviewer #3 (Remarks to the Author):

The manuscript by Schultz et al. entitled "Coordinated regulation of I1061T NPC1 degradation by selective ER autophagy and MARCH6-dependent ERAD" characterizes the degradation mechanisms for NPC1 and several disease-associated NPC1 mutant proteins. The authors conclude that these proteins can be degraded by both the ubiquitin proteasome system and the lysosomes. Upon further examination, they find that although all of these NPC variants are degraded in part by the lysosomes, these proteins appear to use different mechanisms to traffic to the lysosome. Specifically, they propose that the I1061T mutant may use an ER-phagy pathway to reach the lysosome. In my view, the study consists of two parts. Whereas the first part that reports on the pathways involved in degradation of different NPC1 variants lacks novelty (the degradation of NPC1 by ERAD and lysosome has been reported previously), the second part with regard to the involvement of an ER-phagy in degradation of the I1061T mutant but not of other NPC1 proteins is too preliminary. Specifically, there are some technical concerns on several key experiments. Moreover, the study stays at a quite superficial level as no mechanistic insights on why ER-phagy can be so selective is explored.

Specific points:

1. Although the study attempts to address the mechanism of NPC1 turnover, the authors only analyzed the steady state protein levels. Pulse chase or CHX chase is required to convincingly demonstrate that the level of NPC1 is indeed regulated by degradation as opposed to other mechanisms.
2. The overexpression study shown in Figure 5 lacks critical controls to prove that the effect is achieved via the proposed ER-phagy pathway. For example, the authors should mutate the LIM in FAM134B and the mutant should presumably not be able to promote NPC1 degradation.
3. How cells can differentiate between wild type NPC1 and the one with just one amino acid substitution should be explored.
4. In Figure 3, the authors used EndoH and PNGase treatment to discern whether different NPC proteins are transported via different routes. However, given that the authors only analyzed the EndoH sensitivity at the steady state, the results can be subject to alternative interpretations. In one view, it may simply reflect the difference in the rate limiting step during the trafficking process. For example, if the rate limiting step for I1061T turnover is the export of this protein from the ER, it is not surprising to see that the vast majority of the protein is sensitive to EndoH treatment. By contrast, if degradation in the lysosome is

the rate limiting step, most protein would accumulate in a post-Golgi state, which is Endo H resistance. The authors should validate their conclusion using either Brefeldin A or a SAR1 mutant protein to blocks ER to Golgi trafficking.

5. To demonstrate the role of FAM134B in ER-phagy of NPC1 I1061T, the authors only used one siRNA, raising the concern about potential off-target effect.

6. The quality of many immunoblotting data is low (Figure 2c, e, Figure 3b, Figure 4b etc), so is the imaging data.

Reviewer #1:

This manuscript examines the cellular machinery involved in the processing of the disease allele (NPC1 I1061T) most commonly present in Niemann-Pick C1 disease, a neurodegenerative disorder. Previous work in the field demonstrated that the mutant protein is misfolded and degraded by the ERAD. In the present study the authors extend this work by showing that the MARCH6 E3 ubiquitin ligase is responsible for degradation of the I1061T protein. They show that inhibition of lysosome acidification and that inhibition of p97 also increases I1061T protein levels, suggesting that the protein may also be degraded by autophagy. They provide evidence that I1061T protein is degraded by ER-phagy and identify NPC1 I1061T as the first endogenous substrate for this FAM134B-dependent process. The authors show that in the brains of the NPC1 I1061T knockin mouse, FAM134B shifts to the autophagosome, autolysosome, lysosome fractions, and in the cerebella of human NPC1 subjects markers for these compartments, including FAM134B, are elevated.

This study is an important conceptual advance in our understanding of Niemann-Pick C1 disease, specifically the handling of the misfolded I1061T protein in the ER. The in vivo studies in the mouse I1061T model and the human tissue data are consistent with an ER-phagy mechanism and provide strong support for the cell-based observations. Regulation of protein folding in the ER has emerged as a potential target for therapeutic intervention in NPC1 disease because the vast majority of disease is caused by missense mutations, and the majority of these are believed to be misfolding mutants. The discovery of the I1061T protein as a substrate for FAM134B provides new insights into the handling of this mutant in the ER and could represent a point for intervention. NPC1 I1061T protein is apparently the first reported endogenous substrate for FAM134B, but it seems likely that there are other endogenous substrates, which could have implications beyond NPC1 disease. An important question is whether this pathway is operative only for the misfolded mutant NPC1 protein, or whether WT protein is also a substrate. The authors present data suggesting only the mutant protein is being handled by FAM134B, but the data is not convincing (see point 1 below). Previous work (ref 12) showed that ~50% of newly synthesized WT protein does not leave the ER and is degraded. This suggests that the energetics of NPC1 protein folding may involve states that are similar to that of the I1061 protein and thus these states are recognized as misfolded and targeted for degradation. A deeper examination of this mechanism could address this question.

Specific points:

1. An important question is whether the ER-phagy pathway handles both normal and mutant substrates. In response to knockdown of FAM134b, WT NPC1 protein band (fig 4a) clearly is increased compared to untreated (the actin band slightly reduced further yielding a higher ratio), yet the quantification indicates no change. It may be the case that the blot shown is not representative; however, the complicating factor here that the western is measuring total NPC1 protein, the vast majority of which is in the

lysosome, rather than informing with respect to nascent protein. Metabolic labeling coupled with endoglycosidase H treatment studies could definitively answer whether FAM134b knockdown affect degradation of the newly synthesized, ER-associated WT protein. This would be a nice addition to this study.

REPLY:

We thank the reviewer for raising this important question. We have taken the following steps to address this issue:

- We repeated the experiment in question (old Fig 4a; new Fig 5a) additional times, and have updated the western blot images and quantification. We do not detect buildup of wild type NPC1 following knockdown of FAM134B, Beclin1 or p97.
- Additionally, we have repeated experiments in which FAM134B is over-expressed in cells expressing WT or I1061T NPC1 (new Fig 6a). This manipulation was previously shown to increase ER-phagy (Ref 29). Here we now demonstrate a small but significant decrease in WT levels following FAM134B transfection. We believe that this reflects degradation of a pool of misfolded WT NPC1 through ER-phagy, as suggested by the Reviewer. We have added this interpretation to the Results and Discussion sections.
- We performed cycloheximide chase experiments to examine clearance of WT and I1061T NPC1. These experiments show that serum starvation significantly enhances clearance of the I1061T but not WT NPC1 (Fig 4a). Moreover, the enhanced clearance of I1061T due to serum starvation is mitigated by FAM134B knockdown (new Fig 5b).

Collective, these data indicate that much of misfolded I1061T is targeted for degradation by ER-phagy, and that a small fraction of misfolded WT protein is similarly trafficked through this pathway. As noted, this interpretation has been incorporated into the revised Discussion.

2. It seems likely that there are endogenous substrates for the ER-phagy pathway. An excellent candidate would be CFTR, another polytopic transmembrane protein for which the WT protein is largely degraded in the ER. Did the authors examine whether CFTR might also be a substrate for FAM134B?

REPLY:

We agree with the Reviewer that it is likely there are other endogenous substrates for ER-phagy and that CFTR is an excellent candidate. To begin to address this question, we probed lysates from control human fibroblasts treated with non-targeted (NT) siRNA or siFAM134B. We confirmed target knockdown in these samples by qPCR (Supplementary Fig. 4). As shown below, FAM134B knockdown did not significantly alter steady state levels of WT CFTR. Similarly, Houck et al. did not find evidence that deltaF-508 CFTR is degraded in autophagosomes (Ref 28). However, we acknowledge the limitations of our analysis. We believe that this

question will require significant effort to address appropriately using models expressing both WT and mutant CFTR, something best accomplished in future studies in collaboration with an expert in the CFTR field.

3. In addition to I1061T, four compound heterozygote cell lines were also studied, but no rationale is provided for the selection of these mutants. With the exception of the C.1947+5G>C/I1061T line, which only expresses I1061T protein and does not add to the data obtained from the homozygous I1061T cells, the other three express two mutant species, confounding interpretation of the western data. Human cell lines expressing a single NPC1 mutant are commercially available, albeit limited (For example, P1007A compound heterozygote with a null allele is available at Coriell). Alternatively, if the goal were to examine a panel of mutants, the author could consider transient expression in stable lines (Pipalia JLR 2017), though it is not clear that this information is critical for the manuscript. As is, the data seems of limited value and could be removed.

REPLY:

Our goal in this figure panel (Fig 1d) was to establish whether mutant NPC1 levels are responsive to treatment with either proteasome or lysosome inhibitors in multiple independent cell lines that express the mutant protein at endogenous levels. Nonetheless, we understand the Reviewer's perspective. As such, we have chosen to retain in the revised figure data from the C.1947+5G>C/I1061T line, since it corroborates findings in the I1061T homozygous line. Additionally, we show data from a compound heterozygous cell line (P1007A/T1036M) that is studied in subsequent figures. Data from the other two lines we studied have been moved to the supplement and are now presented in Supplementary Fig 1d.

4. Consideration should be given to testing the FAM134B dependence for the NPC1 protein in vivo. This could be achieved by ASO treatment of I1061T/GFP-LC3 mice and monitoring the NPC1 localization in liver tissue. These experiments would also parse out whether the NPC1 protein expressed in the liver and shown to co-localize with the GFP-LC3 marker is due to fusion of autophagosomes to NPC1-containing lysosomes, as suggested by the authors, or a result of FAM134B-dependent trafficking. These studies

would be of particular interest if metabolic labeling studies suggest that WT NPC1 protein is a substrate.

REPLY:

We carefully considered the reviewer's request. Based on our experience in collaboration with Ionis Pharmaceuticals, identification of an optimally active antisense oligonucleotide suitable for in vivo administration is a laborious task that requires screening a large number of ASOs spanning coding and noncoding sequence (in our experience, ~100) followed by in vivo testing. As the scale of this effort is beyond the scope of the current project, we considered generating AAV vectors to express shRNAs to knockdown FAM134B. In consultation with Dr. Beverly Davidson, an expert in this technology, we learned that target knockdown in liver typically occurs slowly after AAV delivery, taking about 2 wks to reach steady state. This prolonged time course, rather than acute knockdown, prompted significant concerns that alternative components of the degradation machinery would be induced in the liver to compensate for FAM134B knockdown. We reasoned that an alternative approach to acutely disrupt autophagy would be more informative.

To accomplish this we prepared sagittal brain slices from littermate WT and I1061T mice and treated them with bafilomycin to inhibit autophagy. Vehicle treatment served as control. As in patient fibroblasts, bafilomycin treatment resulted in an accumulation of p62, our positive control, as well as both WT and I1061T Npc1 protein; these data are now included in new Sup Fig 5.

Lysates were then digested with EndoH or PNGase (see below). We observed a large shift in the migration of WT Npc1 following EndoH digestion and a further small shift after digestion with PNGase. Importantly, treatment of mouse brain lysates with EndoH yielded a more marked shift in WT Npc1 migration than seen in human fibroblasts (for example, compare with figure 3b), while the additional shift resulting from PNGase treatment was quite modest. Analysis performed on I1061T Npc1 yielded a similar pattern, although overall protein levels were very much lower. Notably, the fact that there is a relatively small difference in mobility between EndoH sensitive and resistant species in mouse brain makes it difficult to definitively assess buildup of an EndoH sensitive species using this assay.

Figure: Analysis of Npc1 protein from brain slices after treatment with Baf or vehicle (veh). Lysates were not treated (NT) or digested with EndoH (E) or PNGase

(P). **A.** WT mice (70 µg/lane). **B.** WT or I1061T mice (200 µg/lane). Left panel, short exposure; right panel, long exposure.

Nonetheless, we present evidence using several complementary approaches that NPC1 I1061T is degraded through ER-phagy in vivo. This includes: (1) co-localization of Npc1 I1061T with GFP-LC3 in mouse liver (Fig. 7a); (2) demonstration that the fraction of autophagosomes containing ER as their cargo is significantly increased in livers from I1061T mice (Fig. 7c); demonstration of a shift of FAM134B from ER fractions to those enriched for autophagosomes, autolysosomes, and lysosomes in I1061T brain (Fig. 8a). These data complement our in vitro studies establishing that NPC1 I1061T is degraded through FAM134B-dependent ER-phagy in primary fibroblasts.

Minor points:

1. In several figures, the representative NPC1 bands shown in the westerns do not agree with the quantification. This seems to be the case in fig 4a (see point #1 above) and fig 2d for the siMARCH6-treated normal NPC1 protein. As noted above, the issue may be the background created by the vast excess of lysosome-associated normal protein, swamping out a signal for increased ER-associated protein.

REPLY:

The revised manuscript includes new images for old Fig 4a/new Fig 5a, Fig 2d and Fig 5a. These images are accompanied by updated quantifications. As discussed above, our new data demonstrate a small but significant decrease in WT NPC1 levels following FAM134B over-expression.

2. In WT cells, bafilomycin results in a robust increase in higher MW bands (Fig 1b) and EndoH-resistant NPC1 (Fig 3b), which are not addressed. One interpretation is that inhibition of lysosome acidification is interfering with turnover of fully-glycosylated NPC1 in this compartment. This should be discussed.

REPLY:

This is an excellent point that is now added to the Results.

3. In Fig 6c, what criteria were used to determine that the ER components were contained within autophagosomes?

REPLY:

Criteria used for quantification of TEMs (old Fig 6c/new Fig 7c) are detailed in the Methods. Specifically, autophagic vacuoles (vesicles, 0.5 µm) were classified as

autophagosomes when they met two or more of the following criteria: double membranes (complete or at least partially visible), absence of ribosomes attached to the cytosolic side of the membrane, luminal density similar to cytosol, and identifiable organelles or regions of organelles in their lumen. Vesicles of similar size but with a single membrane (or less than 40% of the membrane visible as double), luminal density lower than the surrounding cytosol or, multiple single membrane-limited vesicles containing light or dense amorphous material were classified as autolysosomes.

Reviewer #2:

In the submitted manuscript “coordinate regulation of I1061T NPC1 degradation by selective ER autophagy and MARCH6-dependent ERAD” the authors analyse the fate of different disease-related NPC1 mutants, with the main focus on I1061T NPC1. The authors showed proteasomal (ERAD) as well as autophagosomal (ER-phagy) participation in the turnover of WT and mutant and analysed the composition of the respective machineries. The manuscript contains novel mechanistic findings behind the Niemann-Pick type C disease and indicates potential therapeutic targets.

While specifically the involvement of ER-phagy in the turnover of disease-related NPC1 mutant proteins is a surprising and therefore important finding for the field, the manuscript in its current form is lacking the maturity and in some parts quality for publication in nature communication.

At this point it is important to mention that the authors completely avoided mentioning the contradiction of their finding of autophagosomal degradation of NPC1 to current literature. This is specifically irritating, as the main authors have published a very nice review on Niemann-Pick type C disease about a year ago, listing publications reporting that autophagy is not involved in NPC1 degradation. In contrast, Schultz et al found autophagy to be the major degradation pathway (80%) of NPC1 mutant I1061T in their setting.

While there are numerous reasons and explanations why this might be the case, it has to be discussed and some key experiments of former studies (usage of the same compound to block autophagy, namely chloroquine and NH₄Cl) have to be repeated in the settings used in this study.

REPLY:

We apologize for not addressing this issue more explicitly in the original draft of the manuscript. Please understand that our previously published review focused on

data in the literature in 2016 showing that chloroquine treatment does not alter NPC1 protein levels. We have made the following changes in the revision:

- We now demonstrate in Supplementary Fig 1b that bafilomycin A1, but not chloroquine, increases levels of I1061T NPC1 in our hands. These data are consistent with published reports, a fact that is explicitly stated and referenced.
- We note that while chloroquine and NH₄Cl are known to raise lysosomal pH by about 1 pH unit (J Cell Biol, 1981,90:665-669; J Cell Biol, 1981, 90:656-664), bafilomycin A1 should bring lysosomal pH closer to neutrality. This may contribute to the greater efficacy of bafilomycin in our assay. However, other downstream effects of bafilomycin treatment cannot be excluded. This comment has been added to the revised Results.

For a reported half-life of 9h for major part of WT and 6.5h for I1061T mutant protein, the authors choose a relatively long treatment time of 24h to detect differences in NPC1 WT and mutant protein levels. This choice and its impact on the experimental outcome (e.g. analysis of different protein pools in comparison to other studies) should be explained to readers.

REPLY:

In the revised manuscript, we state the reported half-life of WT and I1061T NPC1, and indicate that misfolded forms of these proteins likely turnover more rapidly. Additional new data show clearance of these proteins in primary fibroblasts after treatment with cycloheximide (new Fig 4a). Moreover, we provide a time course analysis of effects of bafilomycin (Fig 1c), epoxomicin, and bafilomycin plus epoxomicin on NPC1 levels (new Supplementary Fig 1c).

A control experiment for fitness of cells (e.g. WB of presented lysates for markers of different cell death pathways + respective positive controls) should be included as a supplementary figure to exclude potential artifacts.

REPLY:

We thank the Reviewer for this suggestion. These data are now provided in Supplementary Fig. 1a. We show no significant toxicity following treatment with bafilomycin A1, MG132 or epoxomicin for 24 hr.

Further major and minor points:

Figure 1

Major point:

To understand the contribution of the two degradation pathways and potential

differences for individual protein pools, this time course should be done as well in the presence of MG132 alone and upon double treatment with MG132+Baf.

REPLY:

We now include data showing effects of epoxomicin alone or epoxomicin plus bafilomycin following 4 or 8 hrs treatment. These data are included in Supplementary Fig 1c.

Minor point:

It would be also very informative to have a similar (potentially shorter) time course in the presence of CHX to indicate the contribution of protein synthesis and degradation.

REPLY:

As noted above, we performed cycloheximide chase experiments to examine clearance of WT and I1061T NPC1. These experiments show that serum starvation significantly enhances clearance of the I1061T but not WT NPC1 (new Fig 4a). Moreover, the enhanced clearance of I1061T due to serum starvation is mitigated by FAM134B knockdown (new Fig 5b).

Figure 2

Major point:

Panel a-c) As knockdown of MARCH6 (Fig 2d) has no effect on WT levels, while the WT protein is stabilized upon MG132 treatment (Fig 1a), the same experiments should be done with WT cells.

REPLY:

The experiments in panel 2a examine effects of Sel1 deletion on levels of WT NPC1. Expression of this adapter protein is required for function of the canonical ERAD machinery that includes HRD1. We conclude that loss of this machinery does not alter WT NPC1 levels, nor does it prevent accumulation of WT NPC1 after cells are treated with MG132. We favor the notion that other ERAD components are responsible for triage of WT NPC1 to the proteasome, but the precise identification of this machinery is currently unknown.

Minor point:

Current panel b+c) Since the main focus in this figure lies on the mutant protein, CTRL+FLAG should be excluded from the bar graph (while staying in the WB panel) and intensities should be normalized to I1061T + FLAG / +NT.

REPLY:

As suggested, we now omit the control samples from quantification and normalize intensities to I1061T vector controls.

Figure 3

Major point:

Panel a) Relative intensity should be normalized to I1061T + Vehicle= high to low, less error prone

REPLY:

As suggested, we now normalize intensity to I1061T + vehicle.

Minor point:

Panel b) Presentation of data of individual cell lines would benefit from being on the same gel and thereby presented with same exposure time. As it looks now, e.g. I1061T +MG shows lower NPC1 signal than when treated with Vehicle = contradiction to Fig.1.

REPLY:

Because of differences in protein levels after treatment with bafilomycin or epoxomicin, we are not able to find a single exposure where samples are all in the linear range of exposure and all the bands are demonstrated. Nonetheless, we appreciate this Reviewer's comments and we have re-run all of these experiments in order to obtain higher quality images. Note that our goal in this panel is to examine migration of NPC1 species following enzyme treatment, and we refrain from making any comments on overall protein levels.

Figure 4 (new Figure 5)

Major point:

The whole figure, in specifically quantification of data should be carefully reviewed.

Panel a) In case presented blots are indeed representatives for the performed experiments, one has to doubt the quantification below. By eye and based on a quick ImageJ quantification the levels of WT NPC1 upon silencing of Bec/p97/Fam change 1.17x/1.25x/1.6x and even more profound when taking the loading control into account (1.34x/1.4x/1.9x). These numbers would actually fit to the data presented in Fig1: WT protein levels are increasing upon Baf treatment 2.5x.

REPLY:

We have repeated the experiments in Fig 5a, replaced the blot with a new image, and added additional replicates to the quantification data. We note that while knockdown of FAM134B and other components of the ER-phagy machinery does not alter levels of WT NPC1, we now show that over-expression of FAM134B causes a small decrease of WT protein levels (new Fig 6a), suggesting that a fraction of WT

protein misfolds and is degraded by this pathway. This interpretation has been added to the revised Discussion.

Panel c) Inconsistent NPC1 Levels. I10-Fam/I10-NT is 6.5x in panel a) and less than 1x in panel c)

REPLY:

We replaced the blot in panel 5c with a more representative image.

Panel d) please specify in fig legend appx. number of cells / number of fields quantified for each of the three independent experiments.

REPLY:

We now indicate in the Methods section (under Microscopy) that each field shows $\geq 90\%$ confluence and contains ~ 133 -182 cells. For all experiments where filipin staining was quantified, we examined 16 fields/treatment group/experiment.

Figure 5

Minor point/suggestion:

Even though not significant, WT NPC1 levels seem to be slightly affected by FAM134B overexpression. Specifically when comparing it to the quantification of LIR mutant. This would fit to data presented in Fig1a and WB panels of Fig 4a and could be noted in the text.

REPLY:

As noted above, we performed additional replicates of this experiment and found a small but statistically significant decrease in WT NPC1 levels with FAM134B overexpression. This is now shown in Fig 6a. We suggest that this may reflect clearance of a small fraction of misfolded WT NPC1 through ER-phagy, a notion that has been added to the Discussion. No change in WT or I1061T levels was observed with the LIR mutant (Fig 6c).

Further minor points:

- For clarity CTRL should be replaced by WT in figures and text whenever applicable.

REPLY:

We have sought to clarify our presentation in the revision by stating that control human fibroblasts express WT NPC1. We prefer not to label the control human cells as WT, but would re-label the figures if this were the strong preference of the Reviewer.

- The authors come to the conclusion that accumulated mutant protein is non-

functional (Fig. 3a+b). This is in contrast to published data and could be an interesting point for the discussion.

REPLY:

We do not believe that our findings contrast data in the literature. As noted in the final paragraph of the revised Discussion, the data we present suggest that pharmacological or genetic interventions aimed at restoring mutant NPC1 folding and function will likely need to target components upstream of the degradation pathways studied here. This likely includes machinery that facilitates proper folding of NPC1. Notably, our observations were made in the setting of proteins expressed at endogenous levels. This is different from the important work by Gelsthorpe and colleagues (JBC 283:8229-36, 2008) showing that over-expressed I1061T is functional if properly trafficked. We believe that in this case, transient transfection led to significant over-expression of the I1061T protein. In this setting, the small fraction of mutant protein that properly folded was sufficient to rescue lipid storage.

Reviewer #3:

The manuscript by Schultz et al. entitled "Coordinated regulation of I1061T NPC1 degradation by selective ER autophagy and MARCH6-dependent ERAD" characterizes the degradation mechanisms for NPC1 and several disease-associated NPC1 mutant proteins. The authors conclude that these proteins can be degraded by both the ubiquitin proteasome system and the lysosomes. Upon further examination, they find that although all of these NPC variants are degraded in part by the lysosomes, these proteins appear to use different mechanisms to traffic to the lysosome. Specifically, they propose that the I1061T mutant may use an ER-phagy pathway to reach the lysosome. In my view, the study consists of two parts. Whereas the first part that reports on the pathways involved in degradation of different NPC1 variants lacks novelty (the degradation of NPC1 by ERAD and lysosome has been reported previously), the second part with regard to the involvement of an ER-phagy in degradation of the I1061T mutant but not of other NPC1 proteins is too preliminary. Specifically, there are some technical concerns on several key experiments. Moreover, the study stays at a quite superficial level as no mechanistic insights on why ER-phagy can be so selective is explored.

Specific points:

1. Although the study attempts to address the mechanism of NPC1 turnover, the authors only analyzed the steady state protein levels. Pulse chase or CHX chase is required to convincingly demonstrate that the level of NPC1 is indeed regulated by degradation as opposed to other mechanisms.

REPLY:

We performed cycloheximide chase experiments to examine clearance of WT and I1061T NPC1. These experiments show that serum starvation significantly enhances clearance of the I1061T but not WT NPC1 (new Fig 4a). Moreover, the enhanced clearance of I1061T due to serum starvation is mitigated by FAM134B knockdown (new Fig 5b).

2. The overexpression study shown in Figure 6 lacks critical controls to prove that the effect is achieved via the proposed ER-phagy pathway. For example, the authors should mutate the LIM in FAM134B and the mutant should presumably not be able to promote NPC1 degradation.

REPLY:

We show in Fig 6c that over-expression of FAM134B lacking the LC3 interacting region does not alter expression of either WT or I1061T NPC1.

3. How cells can differentiate between wild type NPC1 and the one with just one amino acid substitution should be explored.

REPLY:

We find this to be an intriguing question that will likely require protein structural analysis to answer. Crystal and cryo-EM structures are currently available for WT NPC1 (Refs 7,8) but not for any missense mutants. These structures place the I1061T mutation in a loop within the luminal domain of NPC1, outside the transmembrane regions that encompass the sterol sensing domain or the N-terminal cholesterol-binding region. Our data, along with data in the literature, support a model in which the I1061T missense mutation leads to protein misfolding in the ER. The precise unfolded conformer triggered by the mutation is currently unknown. To date, we have not obtained any evidence to suggest that the I1061T proteins forms aggregates within the ER lumen. It is our interpretation that the soluble misfolded protein is degraded through ER-phagy. Interestingly, similar misfolding may also occur in a small fraction of WT protein, also leading to its degradation through ER-phagy. These concepts have been added to the revised Discussion.

4. In Figure 3, the authors used EndoH and PNGase treatment to discern whether different NPC proteins are transported via different routes. However, given that the authors only analyzed the EndoH sensitivity at the steady state, the results can be subject to alternative interpretations. In one view, it may simply reflect the difference in the rate limiting step during the trafficking process. For example, if the rate limiting step for I1061T turnover is the export of this protein from the ER, it is not surprising to see that the vast majority of the protein is sensitive to EndoH treatment. By contrast, if degradation in the lysosome is the rate limiting step, most protein would accumulate in a post-Golgi state, which is Endo H resistance. The authors should

validate their conclusion using either Brefeldin A or a SAR1 mutant protein to blocks ER to Golgi trafficking.

REPLY:

As suggested, we now show in Supplementary Fig 1b that treatment with Brefeldin A does not promote accumulation of I1061T NPC1.

5. To demonstrate the role of FAM134B in ER-phagy of NPC1 I1061T, the authors only used one siRNA, raising the concern about potential off-target effect.

REPLY:

Data in Fig 5a show that knockdown of FAM134B with siRNA increases NPC1 I1061T protein levels. For this experiment, we purchased a pool containing four different siRNAs. To determine which ones contribute to I1061T protein accumulation, we subsequently purchased the four individual siRNAs that make up the pool. As now shown in Supplementary Fig 4b, two of the individual siRNA (numbers 2 and 4) are active in triggering accumulation of I1061T protein. The sequences for these siRNAs are included in the revised Methods (page 25).

6. The quality of many immunoblotting data is low, so is the imaging data.

REPLY:

We have replaced many of these panels with higher quality images, including panels 2d, 3b, 5a, 5c and 6a. Additional replicate experiments were performed and quantified.

Reviewers' comments:

Reviewer #1 (Remarks to the Author):

The authors have constructively addressed all of the issues raised in the original review. The new data, in particular the data generated using the brain slices which provide important in vivo insights, has strengthened the manuscript. Overall, the manuscript represents a significant conceptual advance and is an important contribution to the field.

Reviewer #2 (Remarks to the Author):

The manuscript has been improved by the new additions. However, the following issues still need to be addressed:

1. The authors added new data regarding P1007A/T1036M mutant which exhibits a massive accumulation after Baf A1 treatment (page 8 lines 136-137, Fig1d). However, FAM134B knockdown has no effect on it (page 12 line 240-241). The authors should better clarify this point. What is the difference between the two types of mutants? I1061T seems to be affected by FAM134B. Are they located in different sub-compartments of the ER? Do they form different type of aggregates? More discussion is required otherwise this addition (this was asked by the Referee 1) is a bit problematic.
2. Fig 5a. WB panel is incomplete. WB showing reduction of FAM134B must be added as they did for Beclin1 and p97. p67 WB is missing for P1007A/T1036M mutant, not so relevant but just to complete.
3. In vivo data need more discussion. There is a clear accumulation of I1061T aggregates in the liver and a strong presence of autophagosomes with the ER inside (page 15). This makes sense, but FAM134B is not expressed there so clearly some other receptors are involved. What about FAM134A or C or RTN3, SEC62, CCPG1. The authors never mention the possibility that I1061T aggregates may interact also with other ER-phagy receptors, maybe in a cellular or tissue specific manner. SEC62 paper is from 2016, RTN3 paper in 2017 and CCPG1 beginning 2018. In discussion they stress the role of FAM134B in brain so they should speculate some alternatives for the liver or these data better to be removed.
4. Fig8d should be edited. FAM134B is not a transmembrane protein. None of the sub-domains of FAM134B face the ER lumen.

Reviewer #3 (Remarks to the Author):

The key finding reported in this manuscript is that the NPC1 I1061T mutant is degraded in part by lysosomes via an ER-phagy process that is dependent on FAM134B. The role of FAM134B in ER-phagy has been previously reported (Khaminets A et al. 2015). In my opinion, although the finding might be potentially important because mutant NPC1 has been

implicated in Niemann-Pick type C disease, the original study did not go beyond simple description of whether FAM134B or certain genes previously implicated in autophagy (eg. Beclin and p97) are involved in this process. My impression was that the study was preliminary with few mechanistic insights, and in some parts, lacked the quality typically seen in papers published in Nature Communications. In the revised version, the authors do not provide new mechanistic insights, but they did try to address the specific points I raised. I appreciate the efforts they put in to improve this manuscript, but there are still a few issues outstanding.

1. It is important to demonstrate convincingly that NPC1 I1061T is degraded by lysosomes in a FAM134B dependent manner, which is the key finding of this study. In the first draft, the authors only analyzed the steady state expression of NPC1 I1061T by western blot. I believe that authors should perform chase studies to address this deficiency. The authors now include two figures (Figure 4a and 5b), which show quantification of NPC I1061 turnover under three conditions (normal medium, serum starvation and FAM knockdown), but there is no representative gel for these figures. In addition, to substantiate the authors' conclusion, they should also test the turnover of NPC1 I1061 by CHX in the presence of a lysosomal inhibitor (Baf A1 etc.).

2. Another issue related to western blot: Can authors improve the quality of western blots such as Figure 2c? Is this the best looking gel from multiple independent experiments?

3. It is important to demonstrate that the activity of FAM134 in NPC degradation is dependent on the LC3-binding domain, as proposed in Figure 8d. I asked the authors to test that a LC3-binding defective FAM134B mutant fails to promote lysosomal degradation of NPC1 I1061. The data in Figure 6c show that this mutant does not promote NPC I1061T degradation, but there is a major caveat. The authors did not test this mutant side by side with wild type FAM134B. This comparison is critical for the interpretation of this negative result, because unless the two proteins are both properly expressed at a similar level, the authors cannot conclude that the mutant protein is less active than the wild type counterpart. Ideally, CHX chase should be used to test whether these proteins can differentially promote NPC1 degradation.

Reviewer #1 (Remarks to the Author):

The authors have constructively addressed all of the issues raised in the original review. The new data, in particular the data generated using the brain slices which provide important in vivo insights, has strengthened the manuscript. Overall, the manuscript represents a significant conceptual advance and is an important contribution to the field.

Response: We thank the reviewer for the positive comments.

Reviewer #2 (Remarks to the Author):

The manuscript has been improved by the new additions. However, the following issues still need to be addressed:

1. The authors added new data regarding P1007A/T1036M mutant which exhibits a massive accumulation after Baf A1 treatment (page 8 lines 136-137, Fig1d). However, FAM134B knockdown has no effect on it (page 12 line 240-241). The authors should better clarify this point. What is the difference between the two types of mutants? I1061T seems to be affected by FAM134B. Are they located in different sub-compartments of the ER? Do they form different type of aggregates? More discussion is required otherwise this addition (this was asked by the Referee 1) is a bit problematic.

Response: We appreciate this suggestion and have added the following comments to the revised Discussion to address this point: Previous studies demonstrated the importance of FAM134B in promoting selective macroautophagy by targeting perinuclear ER sheets to maintain homeostasis and neuronal function^{29, 43} ... It is interesting to note that the P1007A/T1036M mutant is resistant to effects of FAM134B knockdown (Fig. 5a). It is possible that this mutant misfolds in a different domain of the ER, such as peripheral tubules, and is subject to lysosomal degradation (based on its sensitivity to Baf [Fig. 1d]) but controlled through alternative ER-phagy receptors. This would suggest non-overlapping functional roles of the set of ER-phagy receptors, a topic that warrants further investigation.

2. Fig 5a. WB panel is incomplete. WB showing reduction of FAM134B must be added as they did for Beclin1 and p97. P97 WB is missing for P1007A/T1036M mutant, not so relevant but just to complete.

Response: We updated Supplemental Fig. 4a to include western blot of FAM134B knockdown and Fig 5a to include western blot of p97 for the P1007A/T1036M.

3. In vivo data need more discussion. There is a clear accumulation of I1061T aggregates in the liver and a strong presence of autophagosomes with the ER inside (page 15). This makes sense, but FAM134B is not expressed there so clearly some other receptors are involved. What about FAM134A or C or RTN3, SEC62, CCPG1. The authors never mention the possibility that I1061T aggregates may interact also with other ER-phagy receptors, maybe in a cellular or tissue specific manner. SEC62 paper is from 2016, RTN3 paper in 2017 and CCPG1 beginning 2018. In discussion they stress the role of FAM134B in brain so they should speculate some alternatives for the liver or these data better to be removed.

Response: We thank the reviewer for the suggestion. We have added the following to the revised Discussion: We did not readily detect FAM134B protein in the liver, raising the possibility that other receptors could be driving ER-phagy in this tissue. To initially explore this notion, we used the Human Protein Atlas⁴⁷ to analyze tissue expression of FAM134 family members FAM134B, FAM134A, FAM134C, as well as other recently described ER-phagy receptors SEC62⁴⁴, RTN3⁴⁵, and CCPG1⁴⁶. Consistent with our work here, FAM134B is reportedly expressed at high levels in the brain relative to the liver. High brain expression is mirrored by SEC62, RTN3, and FAM134A. In contrast, both FAM134C and CCPG1 are highly expressed in liver relative to brain. We speculate that misfolded I1061T NPC1 protein may interact with these alternative ER-phagy receptors in a cell or tissue dependent manner.

4. Fig8d should be edited. FAM134B is not a transmembrane protein. None of the sub-domains of FAM134B face the ER lumen.

Response: We apologize for this oversight. Fig. 8d has been modified to show FAM134B does not contain a transmembrane domain.

Reviewer #3 (Remarks to the Author):

The key finding reported in this manuscript is that the NPC1 I1061T mutant is degraded in part by lysosomes via an ER-phagy process that is dependent on FAM134B. The role of FAM134B in ER-phagy has been previously reported (Khaminets A et al. 2015). In my opinion, although the finding might be potentially important because mutant NPC1 has been implicated in Niemann-Pick type C disease, the original study did not go beyond simple description of whether FAM134B or certain genes previously implicated in autophagy (eg. Beclin and p97) are involved in this process. My impression was that the study was preliminary with few mechanistic insights, and in some parts, lacked the

quality typically seen in papers published in Nature Communications. In the revised version, the authors do not provide new mechanistic insights, but they did try to address the specific points I raised. I appreciate the efforts they put in to improve this manuscript, but there are still a few issues outstanding.

Response: We believe that our observations are significant given that (1) I1061T NPC1 is the first endogenous misfolded substrate of FAM134B-dependent ER-phagy; (2) that our report is the first indication that ER-phagy plays a role in clearing the misfolded NPC1 protein in disease; and that (3) our findings have implications for NPC therapy development, particularly the robust, on-going efforts focused on identifying proteostasis regulators.

1. It is important to demonstrate convincingly that NPC1 I1061T is degraded by lysosomes in a FAM134B dependent manner, which is the key finding of this study. In the first draft, the authors only analyzed the steady state expression of NPC1 I1061T by western blot. I believe that authors should perform chase studies to address this deficiency. The authors now include two figures (Figure 4a and 5b), which show quantification of NPC I1061 turnover under three conditions (normal medium, serum starvation and FAM knockdown), but there is no representative gel for these figures. In addition, to substantiate the authors' conclusion, they should also test the turnover of NPC1 I1061 by CHX in the presence of a lysosomal inhibitor (Baf A1 etc.).

Response: In the revised manuscript we now include representative western blots for the CHX chase studies at the critical time point following serum starvation (Supplementary Fig. 3) and FAM134B knockdown (Fig. 5b). As requested, we also test the turnover of NPC1 I1061 by CHX in the presence Baf. As now demonstrated in Supplementary Fig. 3, Baf treatment of I1061T cells during serum starvation prevented NPC1 protein clearance. This further supports our conclusion that I1061T is degraded by autophagy.

2. Another issue related to western blot: Can authors improve the quality of western blots such as Fig. 2c? Is this the best looking gel from multiple independent experiments?

Response: We replaced the western blots in Fig. 2c.

3. It is important to demonstrate that the activity of FAM134 in NPC degradation is dependent on the LC3-binding domain, as proposed in Figure 8d. I asked the authors to test that a LC3-binding defective FAM134B mutant fails to promote lysosomal

degradation of NPC1 I1061. The data in Figure 6c show that this mutant does not promote NPC I1061T degradation, but there is a major caveat. The authors did not test this mutant side by side with wild type FAM134B. This comparison is critical for the interpretation of this negative result, because unless the two proteins are both properly expressed at a similar level, the authors cannot conclude that the mutant protein is less active than the wild type counterpart. Ideally, CHX chase should be used to test whether these proteins can differentially promote NPC1 degradation.

Response: To confirm LIR-FAM-HA over-expression does not alter I1061T NPC1 despite expression differences vs. WT-FAM, the following experiment is now included in Supplementary Fig. 5. I1061T/I1061T fibroblasts were transfected with plasmids as follows:

- 1) 1.5 µg FLAG (control vector)
- 2) 0.75 µg FLAG + 0.75 µg FAM134B-HA
- 3) 1.125 µg FLAG + 0.375 µg FAM134B-HA
- 4) 1.35 µg FLAG + 0.15 µg FAM134B-HA
- 5) 1.5 µg LIR-FAM134B-HA

Cell lysates were collected 48 hours post-transfection and analyzed by western blot. As shown in this new figure, expression of low amounts of WT FAM134B decreased I1061T levels (lanes 3, 4). Notably, FAM134B expression in these lanes *is similar to or lower than* the detected expression of the LIR deletion mutant (lane 5), which did not alter I1061T levels. These observations support the robust and reproducible effect of full-length FAM134B on I1061T levels and the absence of an effect by the LIR deletion mutant.

REVIEWERS' COMMENTS:

Reviewer #3 (Remarks to the Author):

In this revised manuscript by Schultz and colleagues, the authors add the requested western blots. They also did the experiment to compare the activity of wild type FAM134B and the LIM deleted mutant when expressed at a similar level. Overall, they have addressed my questions.